# TIMER-XL: LONG-CONTEXT TRANSFORMERS FOR UNIFIED TIME SERIES FORECASTING

**Yong Liu,**\* **Guo Qin,**\* **Xiangdong Huang, Jianmin Wang, Mingsheng Long**✉
School of Software, BNRist, Tsinghua University, Beijing 100084, China
`{liuyong21,qinguo24}@mails.tsinghua.edu.cn`
`{huangxdong,jimwang,mingsheng}@tsinghua.edu.cn`

## ABSTRACT

We present Timer-XL, a causal Transformer for unified time series forecasting. To uniformly predict multidimensional time series, we generalize next token prediction, predominantly adopted for 1D token sequences, to *multivariate next token prediction*. The paradigm formulates various forecasting tasks as a *long-context* prediction problem. We opt for decoder-only Transformers that capture causal dependencies from varying-length contexts for unified forecasting, making predictions on non-stationary univariate time series, multivariate series with complicated dynamics and correlations, as well as covariate-informed contexts that include exogenous variables. Technically, we propose a universal *TimeAttention* to capture fine-grained intra- and inter-series dependencies of flattened time series tokens (patches), which is further enhanced by deft position embedding for temporal causality and variable equivalence. Timer-XL achieves state-of-the-art performance across task-specific forecasting benchmarks through a unified approach. Based on large-scale pre-training, Timer-XL achieves state-of-the-art zero-shot performance, making it a promising architecture for pre-trained time series models. Code is available at this repository: `https://github.com/thuml/Timer-XL`.

## 1 INTRODUCTION

Transformers have been extensively applied to time series forecasting, becoming the backbone of task-specific models (Zhou et al., 2021; Wu et al., 2021) and pre-trained models (Das et al., 2023). While the majority of prior works have focused on long-term forecasting, reliable predictions are made by considering endogenous variations and exogenous correlations in the context (Box, 2013). Besides, the context length of pre-trained Transformers determines the maximum input and output length during inference. Therefore, long-context Transformers are more versatile than shorter ones, facilitating long-sequence and high-resolution generation (Yin et al., 2023; Wang et al., 2024a).

However, existing Transformers in the time series field crucially encounter the context bottleneck. As shown in Figure 1, unlike Transformers for natural language and vision that learn dependencies among thousands to millions of tokens (Kirillov et al., 2023; OpenAI, 2023), time-series Transformers typically operate around limited contexts of up to hundreds of time series tokens (patches) (Nie et al., 2022). For univariate forecasting, a short-context input leads to insufficient learning of global tendencies, struggling to address non-stationarity in real-world time series (Hyndman, 2018). For multivariate forecasting, increasing research has demonstrated the effectiveness of explicitly capturing intra- and inter-channel dependencies (Zhang & Yan, 2022; Liu et al., 2023; 2024a), highlighting the practical urgency of extending the context length to encompass inter-correlated time series.

Recently, causal Transformers characterized by the decoder-only architecture have become a predominant choice of large language models (Zhao et al., 2023) and garnered increasing attention in the development of large time series models (Rasul et al., 2023; Ansari et al., 2024). Based on contextual flexibility and autoregressive next token prediction, one model can accommodate varying lookback and prediction lengths (Liu et al., 2024b). Therefore, pre-training on longer contexts not only empowers them with the fundamental capability to incorporate more contextual information but

---

\*Equal Contribution

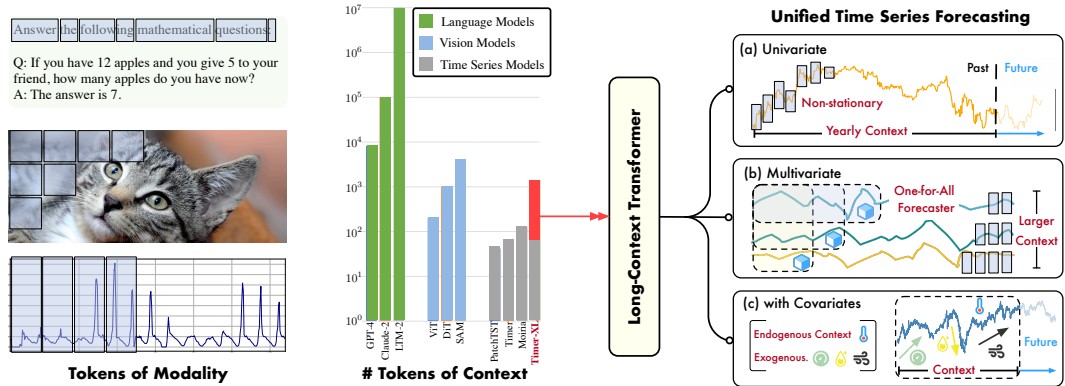

Figure 1: We compare the context length (measured by token number) of Transformers in different modalities and propose Timer-XL that increases the length to thousands of patch tokens. Given the generality across contexts, Timer-XL is a versatile solution for various forecasting tasks.

also enhances the model versatility toward a one-for-all foundation model. Regarding any-variate and any-length time series as one context, previous work (Liu et al., 2024a) has achieved unified modeling on flattened tokens based on noncausal Transformers. However, our empirical results (Figure 3) reveal that encoder-only forecasters may encounter performance degradation in long-context forecasting, while decoder-only Transformers can mitigate this degradation well.

In this work, we generalize the training objective of language modeling to *multivariate next token prediction*, achieving unified time series forecasting that covers tasks in Figure 1 (right). Based on the decoder-only architecture, we propose *TimeAttention* to facilitate Transformers on multidimensional time series, presenting Kronecker-based masking mechanism to train time-series Transformers in a channel-dependent approach. With specialized position embedding for multivariate series, TimeAttention is aware of the chronological order of time points and achieves permutation-equivalence (Zaheer et al., 2017) on variables. We enlarge the context to thousands of patch tokens and achieve state-of-the-art on univariate, multivariate, and covariate-informed forecasting benchmarks. By pre-training on large-scale datasets, we present *Timer-XL* as an extra long version of pre-trained time-series Transformers (Timer) (Liu et al., 2024c), which outperforms recent large models in zero-shot forecasting. Our contributions lie in three aspects:

- We propose multivariate next token prediction and unified time series forecasting, strengthening Transformers with enlarged contexts to make information-complete predictions.

- We introduce TimeAttention, a novel causal self-attention tailored for multidimensional time series, facilitating intra- and inter-series modeling with positional awareness and maintaining causality and scalability of Transformers.

- We propose Timer-XL, a versatile Transformer for one-for-all forecasting, which mitigates performance degradation in long-context time series, achieves state-of-the-art performance in task-specific benchmarks, and presents notable zero-shot performance by pre-training.

## 2 RELATED WORK

Transformers (Vaswani et al., 2017) for time series forecasting have undergone rapid advancements. Initial Transformer-based forecasters primarily focused on *long-term* forecasting (Li et al., 2019; Zhou et al., 2021; Wu et al., 2021; Sun & Zhang, 2024). However, the context length is not growing in pace, which hinders Transformers from making information-complete predictions. Another advancement has focused on multivariate forecasting. Unlike natural language, time series are multidimensional and inherently correlated (Hyndman, 2018). To learn intra- and inter-series dependencies, different tokenization of time-series Transformers has been proposed, including point-wise (Lim et al., 2021), patch-wise (Nie et al., 2022), and variable-wise (Liu et al., 2023) approaches, with deftly tailored architectures (Zhang & Yan, 2022; Wang et al., 2024b). However, few works highlight that multidimensional time series can be uniformly tackled by *long-context* Transformers without architectural

modification. In this work, we leverage causal Transformers, which excel at handling long-context sequences, and unify time series forecasting tasks into multivariate next token prediction.

Recently, time-series Transformers have experienced the evolution from small task-specific models to pre-trained large models (Das et al., 2023; Woo et al., 2024; Ansari et al., 2024). Among them, decoder-only Transformer is predominantly adopted as the backbone of large language models (Touvron et al., 2023; OpenAI, 2023), positioning as a scalable choice for general time series analysis (Liu et al., 2024c). By independently predicting each token with supervision, decoder-only models are also multi-length forecasters (Liu et al., 2024b), avoiding resource-intensive training and lookback-search. However, existing decoder-only Transformers are generally pre-trained in a channel-independent approach, making them inaccessible to inter-series dependencies.

Prior work has employed encoder-only Transformers to capture dependencies of multivariate time series (Liu et al., 2024a). However, our empirical study found that this architecture can be incompatible with causal forecasting, limiting the performance of Transformers. To implement next token prediction and multivariate forecasting in a single Transformer, we renovate the attention module, which disentangles fine-grained token dependencies into variable dependencies and temporal causal masks, capturing intra- and inter-series dependencies with causality and scalability maintained. In Table 1, we list representative time-series Transformers and highlight their differences.

Table 1: Comparison among representative time-series Transformers.

| Model | PatchTST (2022) | iTrans. (2023) | TimeXer (2024b) | UniTST (2024a) | Moirai (2024) | Timer (2024c) | **Timer-XL (Ours)** |
|---|---|---|---|---|---|---|---|
| Intra-Series | ✓ | ✗ | ✓ | ✓ | ✓ | ✓ | ✓ |
| Inter-Series | ✗ | ✓ | ✓ | ✓ | ✓ | ✗ | ✓ |
| Causal Trm. | ✗ | ✗ | ✗ | ✗ | ✗ | ✓ | ✓ |
| Pre-Trained | ✗ | ✗ | ✗ | ✗ | ✓ | ✓ | ✓ |

## 3 APPROACH

In this section, we first introduce a decoder-only Transformer to illustrate the procedure of next token prediction on univariate time series. As an extension, we design *TimeAttention* and propose *Timer-XL* for unified time series forecasting. It is applicable to univariate, multivariate, and covariate-informed scenarios by generalizing the context from 1D sequences to 2D time series.

### 3.1 TIMER

Timer (Liu et al., 2024c) is a time-series Transformer trained by next token prediction (Bengio et al., 2000), which regards single-dimensional time series as non-overlapping patch tokens.

**Next Token Prediction**  Given an univariate time series $\mathbf{X} = \{x_1, \ldots, x_{TP}\}$ of length $TP$, a time series token is defined as $P$ consecutive time points, also termed as the *patch token*:

$$\mathbf{x}_i = \{x_{(i-1)P+1}, \ldots, x_{iP}\} \in \mathbb{R}^P, \ i = 1, \ldots, T. \tag{1}$$

The training objective is to independently predict the next patch token to maximize the likelihood:

$$P(\mathbf{X}) = \prod_{i=1}^{T} p(\mathbf{x}_{i+1}|\mathbf{x}_{\leq i}), \tag{2}$$

which is realized by a decoder-only architecture with the block number $L$ and model dimension $D$:

$$\mathbf{h}_i^0 = \mathbf{W}_e \mathbf{x}_i, \ i = 1, \ldots, T,$$
$$\mathbf{H}^l = \text{TrmBlock}(\mathbf{H}^{l-1}), \ l = 1, \ldots, L, \tag{3}$$
$$\{\hat{\mathbf{x}}_{i+1}\} = \mathbf{H}^L \mathbf{W}_d, \ i = 1, \ldots, T.$$

For simplicity, we omit the block index $l$. Timer adopts $\mathbf{W}_e, \mathbf{W}_d \in \mathbb{R}^{D \times P}$ that independently embed and project the token embeddings as $\mathbf{H} = \{\mathbf{h}_i\} \in \mathbb{R}^{T \times D}$. TrmBlock includes feed-forward network and self-attention with the temporal causal mask $\mathcal{T} \in \mathbb{R}^{T \times T}$. $\mathbf{h}_i \in \mathbb{R}^D$ is the context representation of the previous $i$ tokens. All predicted $\hat{\mathbf{x}}_{i+1}$ are supervised with ground truth via MSE loss.

## 3.2 Generalize 1D Sequences to 2D Time Series

For the enlarged context with the additional dimension, our proposed attention mechanism aims to (1) thoroughly capture intra- and inter-series dependencies and (2) preserve causality within the temporal dimension. Without loss of generality, we illustrate this with the case of multivariate forecasting.

**Multivariate Next Token Prediction**   Given a multivariate time series $\mathbf{X} \in \mathbb{R}^{N \times TP}$ with the number of variables $N$, the time series token $\mathbf{x}_{m,i}$ is defined as the $i$-th patch of the $m$-th variable:

$$\mathbf{x}_{m,i} = \{\mathbf{X}_{m,(i-1)P+1}, \dots, \mathbf{X}_{m,iP}\} \in \mathbb{R}^P, \ m = 1, \dots, N, \ i = 1, \dots, T. \tag{4}$$

The training objective is still to independently predict the next token. Unlike before, each prediction is made based on tokens of the previous time ($\leq i$) from all $N$ variables:

$$P(\mathbf{X}) = \prod_{m=1}^{N} \prod_{i=1}^{T} p(\mathbf{x}_{m,i+1} | \mathbf{x}_{:,\leq i}) = \prod_{m=1}^{N} \prod_{i=1}^{T} p(\mathbf{x}_{m,i+1} | \mathbf{x}_{1,\leq i}, \dots, \mathbf{x}_{N,\leq i}). \tag{5}$$

Compared with Equation 2, the multivariate context length increases from $T$ to $NT$. By contrast, the benefit is that this paradigm learns causal dependencies within each sequence while incorporating exogenous variable correlations from other sequences, making it a universal forecasting paradigm that outperforms channel-independent (Nie et al., 2022) or variable-centric models (Liu et al., 2023).

Technically, we independently apply $\mathbf{W}_e \in \mathbb{R}^{D \times P}$ on each token to obtain patch-wise representation $\mathbf{h}_{m,i} \in \mathbb{R}^D$, which will encompass contextual information from $Ni$ tokens through Transformer blocks and be eventually projected by $\mathbf{W}_d \in \mathbb{R}^{D \times P}$ into the predicted patch token $\hat{\mathbf{x}}_{m,i+1}$.

**Position Embedding**   Position embedding has not been sufficiently explored in time-series Transformers. To avoid inherent permutation-invariance of self-attention, positional embedding is required to reflect the chronological order of tokens on the temporal dimension. As for the variable dimension, shuffling the input order of variables should not affect anything other than the output order of variables. Formally, the processing on multiple variables should be permutation-equivalent (Zaheer et al., 2017).

To meet the above requirements, we adopt RoPE (Su et al., 2024), a widely utilized position embedding on the temporal dimension. For the variable dimension, we use two learnable scalars in each head to keep the permutation-equivalence of variables (Woo et al., 2024). Beyond simply incorporating them together, we provide detailed ablations in Section E.3 to demonstrate the effectiveness:

$$\mathcal{A}_{mn,ij} = \mathbf{h}_{m,i}^{\top} \mathbf{W}_q \mathbf{R}_{\theta,i-j} \mathbf{W}_k^{\top} \mathbf{h}_{n,j} + u \cdot \mathbb{1}(m = n) + v \cdot \mathbb{1}(m \neq n), \tag{6}$$

where $\mathbf{W}_q, \mathbf{W}_k, \mathbf{W}_v \in \mathbb{R}^{D \times d_k}$ and $d_k$ is the dimension of the query, key, and value. $\mathbf{R}_{\theta,t} \in \mathbb{R}^{d_k \times d_k}$ is the rotary matrix with rotation degree $t \cdot \theta$, $\mathbb{1}(\cdot)$ is the indicator function, and $u, v \in \mathbb{R}$ are learnable parameters for the token to distinguish its endogenous and exogenous time series.

**TimeAttention**   In contrast to variable-wise (Liu et al., 2023) and non-causal patch-wise tokens (Nie et al., 2022; Woo et al., 2024), our TimeAttention aims to capture causal patch-wise dependencies within and among all variables. Concretely, we sort patch tokens by flattening their 2D indices into 1D indices in the temporal-first manner, which is illustrated in the upper left of Figure 2. Note that the order of variables does not matter, since Equation 6 guarantees their permutation-equivalence.

We provide an intuitive example to illustrate the causal dependencies within multivariate time series: considering the 2nd token of time series A. To predict its next token, its representation $\mathbf{h}$ should be exactly dependent on the tokens-$\{1, 2, 4, 5\}$. Similarly, we provide all causal dependencies of each token in Figure 12. Based on the visualized attention mask and variable dependencies presented in Figure 2, where all variables are inter-correlated, all token dependencies in $\mathcal{A}$ can be formally disentangled by the Kronecker product into (1) the adjacency matrix of the variable dependency graph $\mathcal{C} \in \mathbb{R}^{N \times N}$ and (2) the causal temporal mask $\mathcal{T} \in \mathbb{R}^{T \times T}$:

$$\mathcal{T}_{i,j} = \begin{cases} 1 & \text{if } j \leq i, \\ 0 & \text{otherwise}, \end{cases} \quad \mathcal{C}_{m,n} = \begin{cases} 1 & \text{if variable } m \text{ is dependent on } n, \\ 0 & \text{otherwise}. \end{cases} \tag{7}$$

Let the Kronecker product $\otimes : (\mathbb{R}^{N \times N}, \mathbb{R}^{T \times T}) \mapsto \mathbb{R}^{NT \times NT}$ take two matrices and produce a block matrix. Consequently, TimeAttention is formulated as follows:

$$\text{TimeAttention}(\mathbf{H}) = \text{Softmax}\left(\frac{\text{Mask}(\mathcal{C} \otimes \mathcal{T}) + \mathcal{A}}{\sqrt{d_k}}\right) \mathbf{H} \mathbf{W}_v, \ \text{Mask}(\mathcal{M}) = \begin{cases} 0 & \text{if } \mathcal{M}_{i,j} = 1, \\ -\infty & \text{if } \mathcal{M}_{i,j} = 0. \end{cases} \tag{8}$$

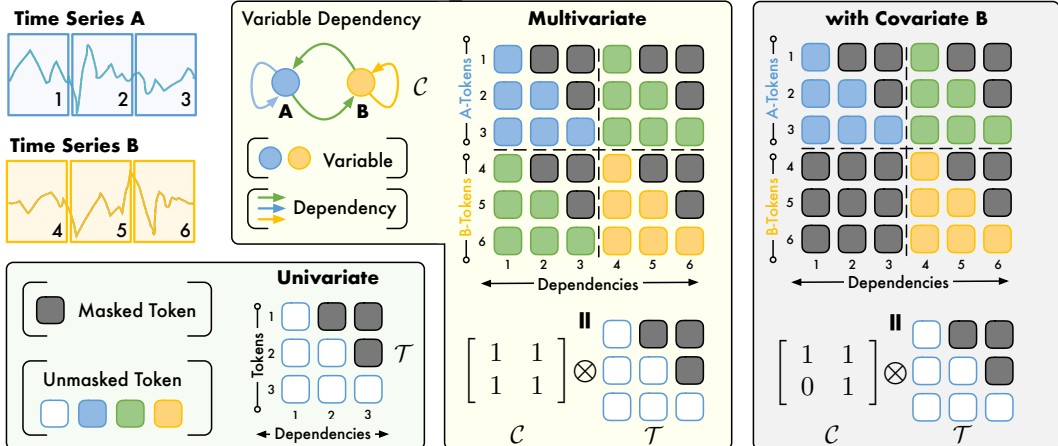

Figure 2: Illustration of TimeAttention. For univariate series, temporal mask $\mathcal{T}$ keeps the causality. Given multivariate patch tokens sorted in a temporal-first order, we adopt the variable dependencies $\mathcal{C}$, an all-one matrix, as the left-operand of Kronecker product, expanding temporal mask to a block matrix, which exactly reflects dependencies of multivariate next token prediction. The formulation is also generalizable to univariate and covariate-informed contexts with pre-defined variable dependency.

Eventually, token representations in $\mathbf{H} = \{\mathbf{h}_{m,i}\} \in \mathbb{R}^{NT \times D}$ will be independently processed by feed-forward network and layer normalization, and fed into the next Transformer block.

**Unified Time Series Forecasting** In multivariate forecasting, the variable dependency forms the complete graph, presenting an all-one matrix $\mathcal{C}$. By generalizing TimeAttention on multiple sequences, Transformers can leverage its length-flexibility to encompass relevant covariates as well. In this case, Timer-XL is adapted in two steps: (1) formulate the customized variable dependency as $\mathcal{C}$ and (2) optimize the model using the supervision of target variables. An example (target-$A$-covariate-$B$) of TimeAttention is illustrated on the right of Figure 2. In a nutshell, we adopt position embeddings for the temporal and variable dimensions. To achieve unified time series forecasting, we flatten 2D time series into a unified context and capture fine-grained causal token dependencies.

## 4 EXPERIMENTS

We conduct evaluations of Timer-XL in three aspects, including (1) supervised training as a task-specific forecaster, (2) large-scale pre-training as a zero-shot forecaster, and (3) assessing the effectiveness of TimeAttention and model efficiency. Given that the long-context forecasting paradigm receives less attention in the community, which can be concealed due to the performance saturation on previous benchmarks (Makridakis et al., 2020; Wu et al., 2022), we established new long-context forecasting benchmarks. Detailed experimental configurations are provided in Appendix B.

### 4.1 UNIVARIATE TIME SERIES FORECASTING

**Setups** Due to the insufficient dataset length when extending contexts in univariate datasets (Makridakis et al., 2020), we adopt multivariate datasets from Liu et al. (2023). Although these datasets are originally multivariate, they aim to be predicted in a univariate approach with the implementation of channel independence. Different from the previous long-term forecasting setting, we focus on reliable prediction based on a long context. Therefore, we fix the prediction horizon and increase the lookback length to monthly and yearly levels. We also establish a long-context univariate benchmark based on the challenging 40-year ECMWF Reanalysis v5 dataset (Hersbach et al., 2020), where yearly contexts are adopted to predict the land-surface temperature of a single site (ERA5-S).

**Results** As shown in Figure 3, the accuracy of univariate prediction can generally be improved by extending the daily context to monthly. We draw a similar conclusion on ERA5 (Table 15), where extending the context consistently helps in the specific model architecture. Notably, Timer-XL with

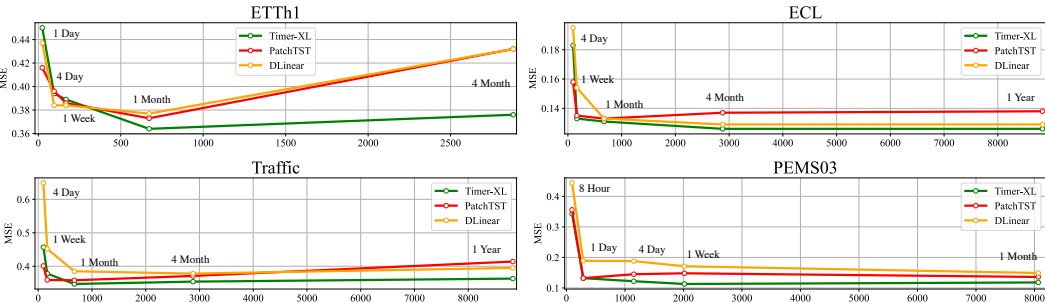

Figure 3: Univariate forecasting (pred-96) of well-acknowledged benchmarks under channel independence (Nie et al., 2022). We increase the lookback length to encompass monthly and yearly contexts.

decoder-only architecture outperforms encoder-only Transformer and linear forecaster in excessively long contexts. Further, we conduct representation analysis in Appendix E.4, revealing that Timer-XL is proficient at adaptively selecting information in vast observations and thus achieves breakthrough performance. It is also noteworthy that the performance of monthly and yearly contexts improves slowly and deteriorates, which may stem from increased noise and training difficulty inherent in data, which leaves a future direction to improve the context efficiency. Table 2 provides results on ERA5-S. Timer-XL consistently outperforms PatchTST on all sites, which can be credited to the maintenance of causality and token-wise supervision in the decoder-only architecture.

**Non-stationary Forecasting**  We delve into widespread non-stationarity in univariate tasks. It is commonly tackled by normalization (Kim et al., 2021) that greatly improves Transformer performance in previous benchmarks. However, we find it may be caused by the insufficient time span and training samples in these datasets. While normalization simplifies learning by aligning series with different means and variances to the same distribution, it limits the model capacity of Transformers, preventing them from learning variations among windows. The by-product can be mode collapse and oversmooth predictions. In Table 2 and Table 16, we evaluate the performance on ERA5 and datasets from Wu et al. (2022), which validates that Timer-XL can achieve better results even without instance normalization.

Table 2: Univariate forecasting (input-3072-pred-96) of ERA5-S, encompassing 117k time points in each station (40-years). We evaluate PatchTST and Timer-XL with and without normalization (Kim et al., 2021). + *Norm.* indicates using the normalization. We train one model for each site separately.

| Station | Beijing | | Hongkong | | London | | New York | | Paris | | Seoul | | Shanghai | | **Average** | |
|---|---|---|---|---|---|---|---|---|---|---|---|---|---|---|---|---|
| Model | MSE | MAE | MSE | MAE | MSE | MAE | MSE | MAE | MSE | MAE | MSE | MAE | MSE | MAE | MSE | MAE |
| PatchTST | 0.0791 | 0.221 | 0.189 | 0.327 | 0.277 | 0.415 | 0.186 | 0.334 | 0.266 | 0.407 | 0.0940 | 0.238 | 0.137 | 0.289 | 0.175 | 0.319 |
| + Norm. | 0.0797 | 0.220 | 0.191 | 0.323 | 0.281 | 0.419 | 0.184 | 0.334 | 0.272 | 0.411 | 0.0914 | 0.233 | 0.136 | 0.287 | 0.176 | 0.319 |
| **Timer-XL** | **0.0739** | **0.210** | **0.179** | **0.316** | **0.262** | **0.404** | 0.182 | **0.327** | **0.254** | **0.399** | 0.0901 | 0.229 | 0.134 | 0.282 | **0.168** | **0.310** |
| + Norm. | 0.0742 | **0.210** | 0.183 | 0.317 | 0.278 | 0.418 | **0.181** | 0.330 | 0.264 | 0.407 | **0.0896** | **0.227** | **0.133** | **0.281** | 0.172 | 0.313 |

## 4.2 MULTIVARIATE TIME SERIES FORECASTING

**Setups**  We follow iTransformer (Liu et al., 2023) to evaluate multivariate forecasting performance. Toward a one-for-all forecaster, we evaluate performance of rolling forecast, that is, we trained one model for all prediction horizons by integrating the previous prediction into the lookback window in the next iteration. We further establish long-context multivariate forecasting benchmarks: ERA5 multi-station land-surface temperature prediction (ERA5-MS), and the global temperature and wind speed forecasting challenge (GTWSF) (Wu et al., 2023), to learn complex temporal dynamics and variable correlations with sufficient training samples.

**Results**  As shown in Tables 3-4 and Figure 4, Timer-XL achieves the best results on both previous and new benchmarks. Essentially, Transformers that explicitly capture inter-series dependencies, such as UniTST (Liu et al., 2024a) and iTransformer, reasonably achieve decent performance in Table 3. Beyond iTransformer, Timer-XL can model fine-grained patch-wise temporal dependencies. With

TimeAttention, Timer-XL outperforms Timer especially on high-dimensional time series (13.2% in ECL and 6.3% in Traffic, with thousands of tokens in the context). Compared with the encoder-only UniTST, decoder-only Transformers excel at generalizing across varying prediction lengths in Table 4.

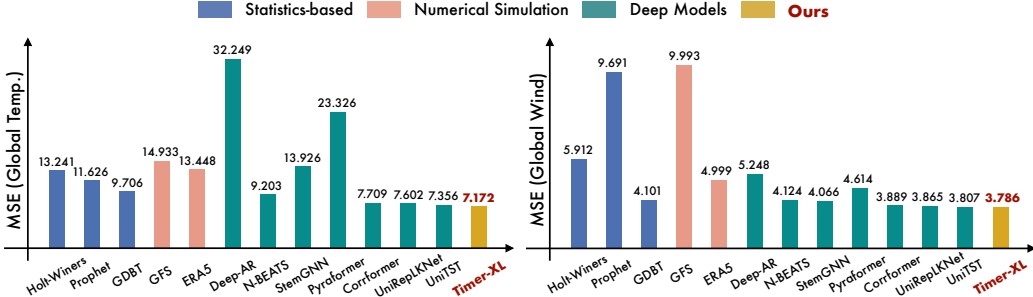

Figure 4: Multivariate forecasting of GTWSF (2-day-pred-1-day), involving 3850 worldwide stations spanning two years. Results of the baseline models are officially reported by Ding et al. (2024).

Table 3: Multivariate forecasting (96-pred-96) of well-acknowledged benchmarks. All models are trained from scratch. Results of baseline models are officially reported by Liu et al. (2023).

| Models | Timer-XL (Ours) | | Timer (2024c) | | UniTST (2024a) | | iTransformer (2023) | | DLinear (2023) | | PatchTST (2022) | | TimesNet (2022) | | Stationary (2022b) | | Autoformer (2021) | |
|---|---|---|---|---|---|---|---|---|---|---|---|---|---|---|---|---|---|---|---|
| Metric | MSE | MAE | MSE | MAE | MSE | MAE | MSE | MAE | MSE | MAE | MSE | MAE | MSE | MAE | MSE | MAE | MSE | MAE |
| ECL | **0.138** | **0.233** | 0.159 | 0.244 | _0.139_ | _0.235_ | 0.148 | 0.240 | 0.197 | 0.282 | 0.181 | 0.270 | 0.168 | 0.272 | 0.169 | 0.273 | 0.201 | 0.317 |
| ETTh1 | **0.381** | **0.399** | 0.386 | 0.401 | 0.385 | 0.402 | 0.386 | 0.405 | 0.386 | _0.400_ | 0.414 | 0.419 | _0.384_ | 0.402 | 0.513 | 0.491 | 0.449 | 0.459 |
| Traffic | **0.387** | **0.260** | 0.413 | _0.265_ | _0.389_ | _0.265_ | 0.395 | 0.268 | 0.650 | 0.396 | 0.462 | 0.295 | 0.593 | 0.321 | 0.612 | 0.338 | 0.613 | 0.388 |
| Weather | **0.165** | **0.209** | 0.176 | 0.215 | **0.165** | _0.210_ | _0.174_ | 0.214 | 0.196 | 0.255 | 0.177 | 0.218 | 0.172 | 0.220 | 0.173 | 0.223 | 0.266 | 0.336 |
| Solar-Energy | **0.200** | **0.229** | 0.204 | 0.234 | _0.203_ | _0.232_ | _0.203_ | 0.237 | 0.290 | 0.378 | 0.234 | 0.286 | 0.250 | 0.292 | 0.215 | 0.249 | 0.884 | 0.711 |

Table 4: Multivariate forecasting (672-pred-{96, 192, 336, 720}) of well-acknowledged benchmarks. We evaluate one-for-all forecasters following Liu et al. (2024b): rolling forecasting for four forecast lengths with one model. Averaged results are reported here and full results are provided in Table 12.

| Models | Timer-XL (Ours) | | Timer (2024c) | | UniTST (2024a) | | iTransformer (2023) | | DLinear (2023) | | PatchTST (2022) | | TimesNet (2022) | | Stationary (2022b) | | Autoformer (2021) | |
|---|---|---|---|---|---|---|---|---|---|---|---|---|---|---|---|---|---|---|---|
| Metric | MSE | MAE | MSE | MAE | MSE | MAE | MSE | MAE | MSE | MAE | MSE | MAE | MSE | MAE | MSE | MAE | MSE | MAE |
| ECL | **0.155** | **0.246** | _0.161_ | _0.251_ | 0.163 | 0.257 | 0.164 | 0.258 | 0.165 | 0.265 | 0.169 | 0.268 | 0.201 | 0.303 | 0.265 | 0.358 | 0.289 | 0.379 |
| ETTh1 | **0.409** | **0.430** | 0.418 | 0.436 | 0.429 | 0.447 | 0.421 | 0.445 | 0.426 | 0.444 | _0.412_ | _0.435_ | 0.495 | 0.491 | 0.505 | 0.513 | 0.517 | 0.528 |
| Traffic | **0.374** | **0.255** | _0.384_ | _0.259_ | 0.385 | 0.265 | _0.384_ | 0.274 | 0.423 | 0.298 | 0.391 | 0.275 | 0.602 | 0.322 | 0.630 | 0.347 | 0.684 | 0.433 |
| Weather | 0.240 | 0.273 | 0.232 | _0.270_ | _0.231_ | 0.272 | 0.266 | 0.291 | 0.239 | 0.291 | **0.226** | **0.268** | 0.264 | 0.293 | 0.308 | 0.329 | 0.435 | 0.455 |
| Solar-Energy | **0.198** | **0.249** | 0.233 | **0.249** | 0.241 | 0.275 | 0.213 | 0.291 | 0.222 | 0.283 | _0.202_ | _0.269_ | 0.213 | 0.295 | 0.254 | 0.315 | 0.265 | 0.325 |

**Ablation Study** Patching (Nie et al., 2022) has been demonstrated as an effective tokenization approach for time series, leading to the boom of Transformers in supervised deep forecasters and large time series models. To better cope with multivariate time series forecasting, we compared typical models on real-world benchmarks to address key questions: (1) whether to conduct explicit inter-series modeling or not (channel independence) and (2) whether to use decoder-only or encoder-only Transformers. The combination presents four Transformers in Table 5, which shows that Timer-XL combines the advantages of explicit inter-series modeling and the decoder-only architecture, which is suitable for multivariate time series forecasting with sufficient training samples.

## 4.3 COVARIATE-INFORMED TIME SERIES FORECASTING

**Setups** For the covariate-informed forecasting, we adopt the well-acknowledged electricity price forecasting (EPF) task (Lago et al., 2021). Each subset contains electricity price as the endogenous variable and two exogenous variables. Therefore, the variable dependency for Timer-XL is formulated

Table 5: Multivariate forecasting (input-3072-pred-96) of ERA5-MS (40 years and 7 stations). We fairly evaluate Transformers that adopt patched time series. *CI.* indicates whether the Transformer uses channel independence (Nie et al., 2022). *Arch.* categorizes them into the encoder-only (E) and decoder-only (D) architectures. Different from ERA5-S in Table 2, we train one model for all sites.

| Station | | | Beijing | | Hongkong | | London | | New York | | Paris | | Seoul | | Shanghai | | **Average** | |
|---|---|---|---|---|---|---|---|---|---|---|---|---|---|---|---|---|---|---|
| Model | CI. | Arch. | MSE | MAE | MSE | MAE | MSE | MAE | MSE | MAE | MSE | MAE | MSE | MAE | MSE | MAE | MSE | MAE |
| PatchTST | Yes | E | 0.0815 | 0.222 | 0.190 | 0.326 | 0.275 | 0.414 | 0.185 | 0.333 | 0.265 | 0.407 | 0.0977 | 0.240 | 0.139 | 0.290 | 0.176 | 0.319 |
| UniTST | No | E | 0.0753 | 0.213 | 0.179 | 0.318 | 0.269 | 0.410 | 0.185 | 0.330 | 0.256 | 0.401 | 0.0901 | 0.230 | 0.135 | 0.284 | 0.170 | 0.312 |
| Timer | Yes | D | **0.0734** | 0.210 | 0.182 | 0.319 | 0.268 | 0.407 | **0.183** | 0.329 | 0.255 | 0.399 | 0.0877 | 0.226 | 0.132 | 0.281 | 0.169 | 0.310 |
| **Timer-XL** | No | D | 0.0736 | **0.209** | **0.174** | **0.309** | **0.263** | **0.404** | 0.182 | **0.327** | **0.252** | **0.396** | 0.0872 | **0.225** | **0.130** | **0.278** | **0.166** | **0.307** |

as $\mathcal{C} = [[1, 1, 1], [0, 1, 0], [0, 0, 1]]$. To investigate whether to learn causal or noncausal patch-wise dependencies in covariates, we implement two versions of Timer-XL: the original one with temporal causal mask $\mathcal{T}$, and the noncausal one with $\mathcal{T}$ replaced by an all-one matrix.

**Results** As shown in Table 6, Timer-XL outperforms state-of-the-art models in covariate-informed tasks. Compared with TimeXer (Wang et al., 2024b), which treats an entire covariate as a token, Timer-XL learns fine-grained patch-wise dependencies. By the noncausal version of Timer-XL, we surprisingly find consistent conclusions with endogenous variables: results will be better if Timer-XL learns causal dependencies within exogenous variables. It again validates that next token prediction that maintains causality has a higher upper limit of performance.

Table 6: Covariate-informed forecasting (168-pred-24) of EPF. We implement two versions of Timer-XL: *Noncausal* indicates that we do not maintain the causality within covariates by replacing temporal causal mask with all-one matrix. Results of baselines are officially reported by Wang et al. (2024b).

| Models | Timer-XL (Ours) | | Timer-XL (Noncausal) | | TimeXer (2024b) | | iTransformer (2023) | | DLinear (2023) | | PatchTST (2022) | | Crossformer (2022) | | TimesNet (2022) | | Autoformer (2021) | |
|---|---|---|---|---|---|---|---|---|---|---|---|---|---|---|---|---|---|---|---|
| Metric | MSE | MAE | MSE | MAE | MSE | MAE | MSE | MAE | MSE | MAE | MSE | MAE | MSE | MAE | MSE | MAE | MSE | MAE |
| NP | **0.234** | **0.262** | 0.237 | 0.265 | 0.238 | 0.268 | 0.265 | 0.300 | 0.309 | 0.321 | 0.267 | 0.284 | 0.245 | 0.289 | 0.250 | 0.289 | 0.402 | 0.398 |
| PJM | 0.089 | **0.187** | 0.092 | 0.188 | **0.088** | 0.188 | 0.097 | 0.197 | 0.108 | 0.215 | 0.106 | 0.209 | 0.149 | 0.198 | 0.097 | 0.195 | 0.168 | 0.267 |
| BE | **0.371** | **0.243** | 0.410 | 0.279 | 0.379 | **0.243** | 0.394 | 0.270 | 0.463 | 0.313 | 0.403 | 0.264 | 0.436 | 0.294 | 0.419 | 0.288 | 0.500 | 0.333 |
| FR | **0.381** | **0.204** | 0.406 | 0.220 | 0.384 | 0.208 | 0.439 | 0.233 | 0.429 | 0.260 | 0.411 | 0.220 | 0.440 | 0.216 | 0.431 | 0.234 | 0.519 | 0.295 |
| DE | **0.434** | **0.415** | 0.435 | **0.415** | 0.440 | 0.418 | 0.479 | 0.443 | 0.520 | 0.463 | 0.461 | 0.432 | 0.540 | 0.423 | 0.502 | 0.446 | 0.674 | 0.544 |
| **Average** | **0.302** | **0.262** | 0.316 | 0.273 | 0.306 | 0.265 | 0.335 | 0.289 | 0.366 | 0.314 | 0.330 | 0.282 | 0.362 | 0.284 | 0.340 | 0.290 | 0.453 | 0.368 |

## 4.4 PRE-TRAINED TIME-SERIES TRANSFORMERS

**Setups** Pre-training enriches time-series Transformers with generalizable forecasting capabilities. The outcome large time series model can cope with widespread challenges of few-shot and zero-shot forecasting. In this section, we conduct univariate pre-training on UTSD (Liu et al., 2024c) and LOTSA (Woo et al., 2024) and evaluate zero-shot performance on benchmarks from Wu et al. (2022). We further conduct large-scale multivariate pre-training on our ERA5-Large dataset, which spans 40 years and encompasses 4920 stations. Subsequently, we evaluate three types of generalization results comparing PatchTST (encoder-only Transformer) and Timer-XL (decoder-only Transformer): pre-training on 80% stations and 80% time span and then forecast on the remaining stations (variable generalization), remaining time span (temporal generalization), and remaining split of time span and stations (variable and temporal generalization). To evaluate the benefit of pre-training with longer context, we compare the zero-shot performance of Timer (2024c) and Timer-XL, where the context length of pre-training is increased from 1440 to 2880.

**Results** We compare generalization performance on ERA5-Large in the middle of Figure 5 (a). Timer-XL achieves better results than PatchTST in all cases, revealing that decoder-only architecture has stronger generalization capability. Figure 5 (b) compares zero-shot performance of two pre-trained Transformers with different context lengths, where Timer-XL outperforms previous Timer on

all benchmark datasets, validating that long-context pre-training enhances large time series models. In Table 7, we provide a comprehensive zero-shot evaluation under a comparable pre-training scale and model size, where Timer-XL achieves notable performance with better sample efficiency. The versatility and scalability make it a promising backbone of foundation models.

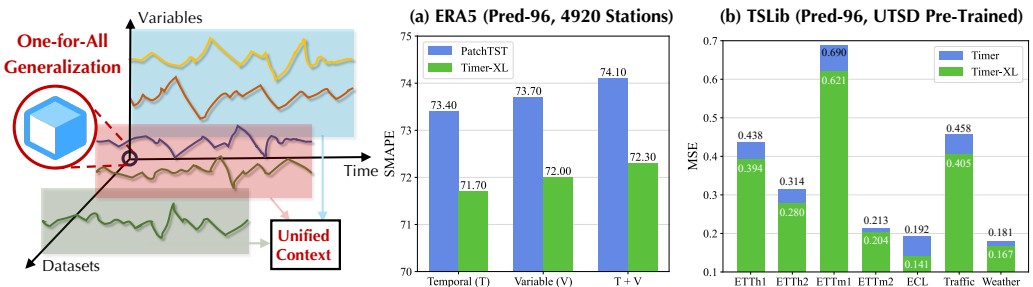

Figure 5: Illustration of one-for-all generalization (left). Based on the contextual flexibility, Timer-XL can predict heterogeneous time series, indicating three directions of generalization shown on the left. We compare performance when generalizing across the time and variables (middle), and zero-shot results across datasets (right), emphasizing the benefit of long-context pre-training.

Table 7: Averaged results of zero-shot forecasting. A lower MSE or MAE indicates a better prediction. Corresponding prediction lengths include $\{96, 192, 336, 720\}$. Full results of all prediction lengths are provided in Table 13. $1^{\text{st}}$ Count represents the number of wins achieved by a model under all prediction lengths and datasets. The detailed configuration of **Timer-XL**$_{Base}$ is provided in Table 11.

| Models | Timer-XL$_{Base}$ (Ours) | | Time-MoE$_{Base}$ (2024) | | Time-MoE$_{Large}$ (2024) | | Time-MoE$_{Ultra}$ (2024) | | Moirai$_{Small}$ (2024) | | Moirai$_{Base}$ (2024) | | Moirai$_{Large}$ (2024) | | TimesFM (2023) | | MOMENT (2024) | | Chronos$_{Base}$ (2024) | | Chronos$_{Large}$ (2024) | |
|---|---|---|---|---|---|---|---|---|---|---|---|---|---|---|---|---|---|---|---|---|---|---|
| Metric | MSE | MAE | MSE | MAE | MSE | MAE | MSE | MAE | MSE | MAE | MSE | MAE | MSE | MAE | MSE | MAE | MSE | MAE | MSE | MAE | MSE | MAE |
| ETTm1 | 0.373 | 0.392 | 0.394 | 0.415 | 0.376 | 0.405 | **0.356** | 0.391 | 0.436 | 0.410 | 0.406 | **0.385** | 0.422 | 0.391 | 0.433 | 0.418 | 0.670 | 0.536 | 0.645 | 0.500 | 0.555 | 0.465 |
| ETTm2 | **0.273** | **0.336** | 0.317 | 0.365 | 0.316 | 0.361 | 0.288 | 0.344 | 0.307 | 0.347 | 0.311 | 0.337 | 0.329 | 0.343 | 0.328 | 0.346 | 0.316 | 0.365 | 0.310 | 0.350 | 0.295 | 0.338 |
| ETTh1 | 0.404 | **0.417** | 0.400 | 0.424 | **0.394** | 0.419 | 0.412 | 0.426 | 0.428 | 0.427 | 0.417 | 0.419 | 0.480 | 0.439 | 0.473 | 0.443 | 0.683 | 0.566 | 0.591 | 0.468 | 0.588 | 0.466 |
| ETTh2 | **0.347** | 0.388 | 0.366 | 0.404 | 0.405 | 0.415 | 0.371 | 0.399 | 0.361 | 0.384 | 0.362 | 0.382 | 0.367 | **0.377** | 0.392 | 0.406 | 0.361 | 0.409 | 0.405 | 0.410 | 0.455 | 0.427 |
| ECL | **0.174** | 0.278 | - | - | - | - | - | - | 0.218 | 0.303 | 0.187 | 0.274 | 0.186 | **0.270** | - | - | 0.765 | 0.686 | 0.214 | 0.278 | 0.204 | 0.273 |
| Weather | **0.256** | 0.294 | 0.265 | 0.297 | 0.270 | 0.300 | **0.256** | 0.288 | 0.275 | 0.286 | 0.287 | 0.281 | 0.264 | **0.273** | - | - | 0.294 | 0.326 | 0.292 | 0.315 | 0.279 | 0.306 |
| $1^{\text{st}}$ Count | **15** | **10** | 2 | 1 | 3 | 0 | 10 | 7 | 0 | 0 | 0 | 5 | 1 | **10** | 0 | 1 | 2 | 0 | 0 | 0 | 0 | 2 |

∗ Dataset for pre-training is not evaluated on corresponding models, which is denoted by a dash (−).

∗ Traffic from (PEMS) is generally used during the pre-training of large models and thus not evaluated here.

∗ Our model checkpoint is available at https://huggingface.co/thuml/timer-base-84m.

## 4.5 MODEL ANALYSIS

**Model Efficiency** To evaluate the model efficiency of Timer-XL with respect to the context length, it is essential to recognize the distinct characteristics of time series data compared to 1D sequences. Unlike natural language, the time series modality is characterized by the variable number $N$ and the input length. We adopt two representative multivariate datasets with different $N$, and provide the memory footprint and training speed under gradually prolonged input. We evaluate typical approaches to handle multivariate series: (1) Timer-XL and Moiria that adopt channel dependence; (2) Timer that adopts channel independence. Intuitively, the complexity of the first type is $\mathcal{O}(N^2 T^2)$ while the complexity of self-attention under channel independence is $\mathcal{O}(NT^2)$. However, results shown in Figure 6 reveal that measured overheads of Timer-XL is much less than $N$ times of Timer.

Since the previous analysis of model efficiency on time-series Transformer predominantly focuses on self-attention on 1D time series, we initially present a theoretical derivation of the computational complexity of Transformers on 2D time series, including the parameter counts, memory footprint, and FLOPs in Table 8. We find that other parts of Transformers, such as feed-forward network, have a complexity of $\mathcal{O}(NT)$ no matter which approach is adopted to handle multivariate time series. They also account for dominant overheads in existing benchmarks since the context length is not large enough, confirming our empirical results. Further, we introduce FlashAttention (Dao et al., 2022) to

improve the model efficiency, which is computationally equivalent and reduces the overall memory footprint of Timer-XL to $\mathcal{O}(NT)$ without affecting performance.

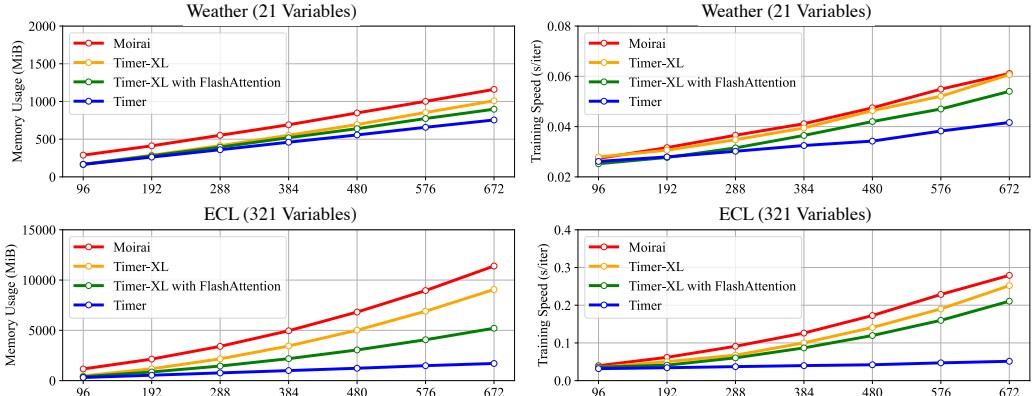

Figure 6: Efficiency analysis. We compare representative time-series Transformers on multivariate datasets with variable numbers ranging from ten to hundred and increase the lookback length.

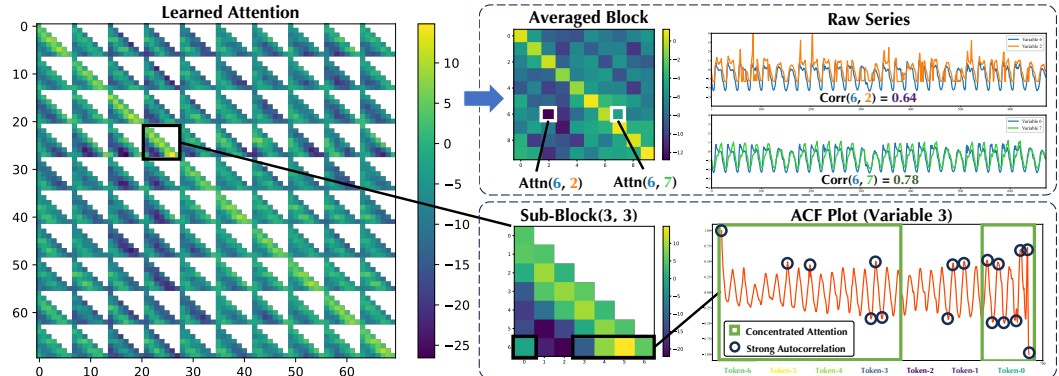

Figure 7: Visualization of TimeAttention. It is from the first sample of a length 672 in the test split of Traffic. We visualize the last 10 variables with each contains 7 tokens. We present auto-correlation function plot. Auto-correlation can be reflected by the distribution of attention scores (bottom right). We average TimeAttention across sub-blocks, which indicates Pearson correlations (upper right).

**Representation Analysis** In addition to the enhanced performance, fine-grained token dependencies offer improved interpretability. We present a showcase visualization from Traffic in Figure 7. It is observed that sub-matrices along the diagonal generally receive greater attention, which reasonably reveals predominant dependencies within the endogenous variable. By zooming in a sub-block that corresponds to Variable-3, we observe that the attention distribution of the last row can indicate certain strong dependencies among patch tokens. This observation is also supported by the auto-correlation function plot (ACF), which reveals auto-correlations with certain lags and thus the model pays special attention to these tokens. Furthermore, we average each sub-matrix into one scalar. The outcome matrix can also illustrate Pearson correlations presented in the raw data.

## 5 Conclusion and Future Work

In this paper, we emphasize the efficacy of causal Transformers in the forecasting of long-context time series. To facilitate long-context Transformers on diverse tasks, we propose multivariate next token prediction, a novel paradigm to predict multidimensional series with covariates. We present Timer-XL enhanced by TimeAttention as an extra-long version of pre-trained time-series Transformers. It simultaneously captures temporal dynamics and variable correlations by enhanced self-attention. In addition to achieving state-of-the-art performance on extensive benchmarks, we establish challenging benchmarks for long-context forecasting. By pre-training on large-scale heterogeneous time series, Timer-XL demonstrates notable zero-shot performance as a large time-series model. In the future, we will improve computational efficiency and build large domain-specific models with Timer-XL.

## ACKNOWLEDGMENTS

This work was supported by the National Natural Science Foundation of China (U2342217 and 62021002), the BNRist Project, and the National Engineering Research Center for Big Data Software.

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

## A    PROOF OF MODEL EFFICIENCY

### A.1    SETUPS

Given an input univariate time series divided into $T$ tokens according to the patch size $P$, which is fed into the vanilla Transformer. The training objective is to predict the next token of $P$ time points. We will generalize the derivation from 1D sequences to 2D time series based on different approaches to handle multivariate data with the variable number $N$. We adopt the same denotations as before: Transformer consists of $L$ blocks with model dimension $D$. The multi-head attention mechanism has $H$ heads, each with a dimension of $d_k$ for query, key, and value, and $d_k = \frac{D}{H}$. The intermediate dimension of feed-forward network is set as $D_{\text{ff}} = \alpha D$. The results are summarized in Table 8, we provide the detailed proof in the following sections.

Table 8: Parameters count and computational complexity of Transformers for multivariate time series.

| Metric | Type | Count | Complexity |
|---|---|---|---|
| FLOPs (Training Speed) | Channel Independence | $12\big(PDNT + L(D+H)NT^2 + (2+\alpha)LD^2NT\big)$ | $\mathcal{O}\big(LDNT(D+T)\big)$ |
| | Channel Dependence | $12\big(PDNT + L(D+H)N^2T^2 + (2+\alpha)LD^2NT\big)$ | $\mathcal{O}\big(LDNT(D+NT)\big)$ |
| Parameters | Encoder-Only | $(4+2\alpha)LD^2 + 4LD + (1+T)PD$ | $\mathcal{O}\big(LD^2\big)$ |
| | Decoder-Only | $(4+2\alpha)LD^2 + 4LD + 2PD$ | $\mathcal{O}\big(LD^2\big)$ |
| Memory Footprint | Self-Attention | $4(D+P)NT + (32+8\alpha)LDNT + 4LHN^2T^2$ | $\mathcal{O}\big(LHN^2T^2\big)$ |
| | FlashAttention | $4(D+P)NT + (32+8\alpha)LDNT$ | $\mathcal{O}\big(LDNT\big)$ |

∗ $L$ is the block number of Transformers. $D$ is the dimension of embeddings (the hidden dimension of FFN $D_{\text{ff}}$ is set as $\alpha D$). $H$ is the head number and the dimension of query, key, and value $d_k = D/H$. The overhead is to train on a multivariate time series ($N$-variables and $TP$ time points) with patch token length $P$ and context length $T$. Set $N = 1$ for training on univariate time series.

### A.2    FLOPs

As a preliminary, the multiplication between matrix $\mathbf{A} \in \mathbb{R}^{n \times m}$ and matrix $\mathbf{C} \in \mathbb{R}^{m \times p}$ requires $mnp$ multiplications and $mnp$ additions, resulting in $2mnp$ floating-point operations. Given batched matrices $\mathbf{A} \in \mathbb{R}^{B \times n \times m}$ and $\mathbf{C} \in \mathbb{R}^{B \times m \times p}$ , $B$ times matrix multiplications will be performed. It is evident that the batch size is a linear multiplier. Thus, we first omit $B$ to calculate the operations of dealing with one univariate series, and then we will reintroduce it to analyze channel independence.

The computational cost of Transformers can be primarily categorized into two types: (1) multi-head attention calculation and (2) linear transformations. In contrast, the operations of layer normalization, residual connection, activation functions, and position embedding with the complexity of $\mathcal{O}(TD)$ are less significant. Therefore, we derive the computational complexity mainly with respect to the above two types by delving into the forwarding process of one univariate series.

**Patch Embedding**    The tokenized time series $\{\mathbf{x}_i\} \in \mathbb{R}^{T \times P}$ is mapped into the embedding space through the patch-wise embedding $\mathbf{W}_e \in \mathbb{R}^{D \times P}$ , resulting in $2PDT$ operations.

**Self-Attention**    The calculation of self-attention begins with the computation of query, key and value by multiplying the patch embeddings with matrices $\mathbf{W}_q, \mathbf{W}_k, \mathbf{W}_v \in \mathbb{R}^{D \times d_k}$ respectively in $H$ heads, which incurs a computational cost of $6HDd_kT = 6D^2T$ and yields $\mathbf{Q}, \mathbf{K}, \mathbf{V} \in \mathbb{R}^{H \times T \times d_k}$. Next, the dot product $\mathbf{Q}\mathbf{K}^\top \in \mathbb{R}^{H \times T \times T}$ is conducted in each head, leading to $2Hd_kT^2 = 2DT^2$ operations. Following this, the Pre-Softmax map is divided by $\sqrt{d_k}$ and processed through $\mathrm{Softmax}$, which includes exponentiation, summation, and normalization of each element, resulting in $4HT^2$ operations. The subsequent multiplication with $\mathbf{V}$ incurs $2Hd_kT^2 = 2DT^2$ operations. Finally, multiple heads are concatenated and multiplied by $\mathbf{W}_o \in \mathbb{R}^{D \times D}$, contributing $2D^2T$ operations.

**Feed-Forward Network** It first projects the token representations into the dimension of $D_{ff}$ and subsequently projects it back to the dimension $D$, resulting in a total operations of $4\alpha D^2 T$.

**Patch Projection** For encoder-only models, all token representations are flattened and mapped directly to $P$ time points by $\mathbf{W}_d \in \mathbb{R}^{TD \times P}$. In contrast, token-wise projector $\mathbf{W}_d \in \mathbb{R}^{D \times P}$ in decoder-only models independently map each token to the predicted next token. In both cases, the number of operations is $2PDT$, but the token-wise projector will result in a smaller parameter count.

The forwarding operations in $L$-layers Transformer is $4PDT + 4L(D + H)T^2 + (8 + 4\alpha)LD^2 T$ in sum. Considering that the majority of operations in Transformers are binary operations (e.g., matrix multiplications), the gradients for both matrices are computed separately. As a result, the number of operations in backpropagation is the twice of forwarding. Therefore, the total operations of training a Transformer on a univariate series consisting of $T$ patches, each of length $P$, is derived as:

$$f(T) = 12PDT + 12L(D + H)T^2 + (24 + 12\alpha)LD^2 T.$$

We plug typical hyperparameters in the current time-series Transformers and forecasting benchmarks: $D = 512, H = 8, L = 4, \alpha = 4, T = 7$, and $P = 96$, we obtain that:

$$f(T) = 24960T^2 + 76087296T \propto 3.28 * 10^{-4}T^2 + T.$$

Due to the prevalence of short contexts in the time series field, where $T \ll D$ leads to a significant coefficient in $\mathcal{O}(T)$, we find the primary computational burden of time-series Transformer lies in linear transformations with $\mathcal{O}(T)$, rather than in multi-head self-attention with the $\mathcal{O}(T^2)$ complexity.

For multivariate series with $N$ variables, FLOPs is influenced by the handling of multivariate data. When adopting channel independence (Timer and PatchTST), $N$ can be regarded as the batch size $B$:

$$Nf(T) = 12PDNT + 12L(D + H)NT^2 + (24 + 12\alpha)LD^2 NT. \tag{9}$$

For models that capture fine-grained intra- and inter-series dependencies (Timer-XL and UniTST) in multivariate series, $N$ is reflected as the enlarged number of tokens:

$$f(NT) = 12PDNT + 12L(D + H)N^2 T^2 + (24 + 12\alpha)LD^2 NT. \tag{10}$$

Notably, FLOPs is not entirely equivalent to actual runtime. While FlashAttention increases the overall FLOPs due to its recomputation process, it reduces the number of memory reads and writes. Given that on GPUs, computation is significantly faster than memory access, using FlashAttention can actually lead to further improvements in runtime performance.

## A.3 PARAMETER COUNT

From the above analysis, we observe that the parameter count of Transformers includes the following:

**Patch Embedding** $\mathbf{W}_e \in \mathbb{R}^{D \times P}$ to obtain patch embeddings.

**Self-Attention** $\mathbf{W}_q, \mathbf{W}_k, \mathbf{W}_v \in \mathbb{R}^{D \times d_k}$ of $H$ heads and $\mathbf{W}_o \in \mathbb{R}^{D \times D}$ for all heads.

**Feed-Forward Network** $\mathbf{W}_{\text{ffn1}}, \mathbf{W}_{\text{ffn2}} \in \mathbb{R}^{D \times D_{\text{ff}}}$ in feed-forward network.

**Layer Normalization** It contains the weight $\mathbf{W} \in \mathbb{R}^D$ and the bias $\mathbf{b} \in \mathbb{R}^D$. Every Transformer block includes two normalizations after multi-head attention and feed-forward network respectively.

**Patch Projection** $\mathbf{W}_d \in \mathbb{R}^{TD \times P}$ in flatten head and $\mathbf{W}_d \in \mathbb{R}^{D \times P}$ in token-wise projection.

In sum, the total count of parameters in time-series Transformers can be expressed as:

$$\text{Parameter Count} = \begin{cases} (4 + 2\alpha)LD^2 + 4LD + (1 + T)PD, & \text{using flatten head,} \\ (4 + 2\alpha)LD^2 + 4LD + 2PD, & \text{using token-wise projection.} \end{cases} \tag{11}$$

## A.4 MEMORY FOOTPRINT

The memory footprint during training can be primarily categorized into three parts: activation values stored for backpropagation, model parameters, and optimizer parameters.

Regardless of other precision types (e.g., FP16), model parameters and gradients are typically stored as 32-bit floating-point numbers, with each parameter occupying 4 bytes of memory. For time-series Transformers, memory footprint of activation values is given as follows:

**Patch Embedding**    Gradient computation for $\mathbf{W}_e$ preserves its input $\{\mathbf{x}_i\} \in \mathbb{R}^{T \times P}$ of $4PT$ bytes.

**Self-Attention**    Gradient calculation for $\mathbf{W}_q, \mathbf{W}_k, \mathbf{W}_v \in \mathbb{R}^{D \times d_k}$ requires their inputs $\mathbf{H} \in \mathbb{R}^{T \times D}$, amounting to a total of $4DT$ bytes. The dot product for attention map also needs to store $\mathbf{Q}, \mathbf{K}, \mathbf{V} \in \mathbb{R}^{H \times T \times d_k}$, which collectively require a total of $12DT$ bytes of memory. Gradient computation of $\mathbf{W}_o \in \mathbb{R}^{D \times D}$ necessitates the concatenated multi-head attention representations $\mathbf{H} \in \mathbb{R}^{T \times D}$, which occupies $4DT$ bytes. If memory-efficient attention mechanisms like FlashAttention (Dao et al., 2022) is not applied, the outcome $\mathbf{Q}\mathbf{K}^\top$ will be stored and occupy $4HT^2$ bytes. Instead, if FlashAttention is adopted, the storage overhead can be avoided.

**Feed-Forward Network**    ReLU activation function is typically employed in this module. The input $\mathbf{H} \in \mathbb{R}^{T \times D}$ must be retained, requiring a total of $4DT$ bytes. Additionally, the product $\mathbf{W}_{\text{ffn1}}\mathbf{H}$ also needs to be stored, amounting to $4D_{\text{ff}}T$ bytes. Similarly, the output activations of ReLU, which serve as the input for subsequent linear transformations, necessitate another $4D_{\text{ff}}T$ bytes.

**Layer Normalization**    Each block of Transformer encompasses two layer normalizations, with each normalization retaining its input, resulting in the memory requirement of $8DT$ bytes.

**Patch Projection**    To perform backpropagation for $W_d \in \mathbb{R}^{D \times P}$, it is necessary to retain its input $\mathbf{H} \in \mathbb{R}^{T \times D}$, resulting in a total memory requirement of $4DT$ bytes.

The formula for the total activation values of the entire model occupying GPU memory is as follows:

$$\text{Memory Footprint} = \begin{cases} 4(D+P)T + (32 + 8\alpha)LDT + 4LHT^2, & \text{w/o FlashAttention}, \\ 4(D+P)T + (32 + 8\alpha)LDT, & \text{with FlashAttention}. \end{cases} \quad (12)$$

The derived occupancy of activation values increases proportionally with the batch size $B$. For multivariate series, $N$ can be used as a multiplier in channel independence. For channel independence models, we can substitute $T$ with $NT$ as before. The total memory footprint is the sum of activation values and parameters of model and optimizer, which are proportional to the parameter count derived in Equation 11. Due to the limited model size in the time series field, the memory consumption of parameters is minimal and can be considered negligible in practice. Therefore, the overall memory footprint can be predominantly determined by the occupied memory of activation values.

## B  EXPERIMENTAL DETAILS

### B.1  DATASETS

We conduct experiments on well-acknowledged benchmarks to evaluate performance of the proposed Timer-XL, which includes (1) ETT (Zhou et al., 2021) contains 7 factors of electricity transformers from July 2016 to July 2018, which is recorded every hour or 15 minutes. (2) Weather (Wu et al., 2021) includes 21 meteorological factors collected every 10 minutes from the Max Planck Biogeochemistry Institute Weather Station in 2020. (3) ECL (Wu et al., 2021) records the hourly electricity consumption data of 321 clients. (4) Traffic (Wu et al., 2021) collects hourly road occupancy rates measured by 862 sensors on the San Francisco Bay area highways from January 2015 to December 2016. (5) Solar-Energy (Lai et al., 2018) records the solar power production of 137 PV plants in 2006, which are sampled every 10 minutes. (7) PEMS (Liu et al., 2022a) contains records from the public traffic network in California collected in 5-minute time windows. (8) EPF (Lago et al., 2021) includes five subsets that span six years. Each contains the electricity price as the endogenous variable to be predicted and two exogenous variables of the day-ahead electricity markets. (9) GTWSF (Wu et al.,

2023) is a dataset collected from the National Centers for Environmental Information (NCEI). This large-scale collection contains hourly averaged wind speed and temperature data from 3850 stations with different geographical scales and densities each, spanning from 2019 to 2021. (10) UTSD (Liu et al., 2024c) is a multi-domain time series dataset, which includes seven domains with a hierarchy of four volumes. We adopt the largest volume that encompasses 1 billion time points for pre-training.

We further establish challenging forecasting benchmarks based on the ECMWF Reanalysis v5 (ERA5) dataset (Hersbach et al., 2020) to prevent potential overfitting and performance saturation of deep forecasters in existing benchmarks. Concretely, ERA5 is the fifth generation ECMWF atmospheric reanalysis of the global climate covering the period from January 1940 to the present, which provides hourly estimates of a large number of atmospheric, land, and oceanic climate variables, and includes information about uncertainties for all variables at reduced spatial and temporal resolutions. Due to its pattern sufficiency of temporal dynamics and variable correlations, we could establish practical benchmarks to thoroughly evaluate the performance for univariate and multivariate forecasting, as well as adopt it for large-scale pre-training to develop domain-specific large time series models.

Our datasets are constructed as follows:

- **ERA5-S**: To establish a realistic univariate forecasting benchmark, we start from the basic principle of forecastability and make the prediction on sufficient lookback lengths. Instead of the short time span of training in previous benchmarks (generally no more than 2 years), we curated a three-hour frequency dataset spanning 40 years (January 1979 to December 2018) from ERA5, encompassing 116880 time points. In order to prevent overfitting on a single time series, we selected worldwide stations to form seven subsets.

- **ERA5-MS**: Each univariate series of ERA5-S provides partial observations governed by the spatio–temporal global weather system. Since discovering the global spatio-temporal correlations presents a fundamental challenge in meteorology, we convert ERA5-S into ERA5-MS by using seven subsets as a challenging multivariate forecasting benchmark. Based on the average results in Tables 2 and 5, we can validate the existence of multi-station correlations among selected stations, which have enhanced the average prediction accuracy.

- **ERA5-Large**: To explore the pure data-driven approach to build domain-specific large time series models, we further expanded the number of stations as ERA5-Large, a dataset that evenly covers meteorological 4920 worldwide stations and spans 40 years. We establish the dataset for pre-training, which is expected to generalize across the time (train on the past observations and generalize to the future) and across stations (train on partial stations and generalize to other unseen stations). The total number of time points is around half a billion.

We follow the same data processing and train-validation-test split protocol used in TimesNet (Wu et al., 2022), where the train, validation, and test datasets are divided according to chronological order to prevent data leakage. Detailed dataset descriptions and prediction settings are provided in Table 9.

## B.2 BASELINE MODELS

We aim to present Timer-XL as a foundation model for unified time series forecasting. We thoroughly include well-acknowledged and advanced models in each forecasting task. For univariate time series forecasting, we compare Timer-XL with PatchTST (Nie et al., 2022) under channel independence. For multivariate time series prediction, we report official results from Liu et al. (2023; 2024b); Ding et al. (2024), including UniRepLKNet (2024), iTransformer (2023), Corrformer (2023), DLinear (2023), TimesNet (2022), Non-stationary Transformer (2022b), Pyraformer (2021), Autoformer (2021), StemGNN (2020), DeepAR (2020), and N-BEATS (2019). We further reproduce the performance of related Transformers: Timer (2024c) and UniTST (2024a) based on their official repositories. For covariate-informed time series forecasting, we report the official results of TimeXer (2024b). For zero-shot forecasting, we follow Liu et al. (2024c) that predicts future length-96 windows in well-acknowledged datasets. Totally, more than 20 baselines are included for a complete comparison.

## B.3 IMPLEMENTATION DETAILS

All the experiments are implemented by PyTorch (Paszke et al., 2019) on NVIDIA A100 Tensor Core GPUs. We employ the Adam optimizer (Kingma & Ba, 2014) and MSE loss for model optimization.

Table 9: Dataset descriptions. *Dim.* denotes the number of variables (For univariate forecasting, we adopt channel independence (Nie et al., 2022) or train separate models on each variable). *Dataset Length* denotes the number of time points in the (train, validation, test) splits.

| Tasks | Dataset | Dim. | Training Setting | Dataset Length | Information (Frequency) |
|---|---|---|---|---|---|
| Univariate Forecasting | ETTh1 | 7 | {24, 96, 168, 672, 2880}→96 | (8545, 2881, 2881) | Electricity (Hourly) |
| | ECL | 321 | {24, 96, 168, 672, 2880, 8832}→96 | (18317, 2633, 5261) | Electricity (Hourly) |
| | Traffic | 862 | {24, 96, 168, 672, 2880, 8832}→96 | (12185, 1757, 3509) | Transportation (Hourly) |
| | PEMS03 | 358 | {96, 288, 1152, 2016, 8064}→96 | (15617, 5135, 5135) | Transportation (5 mins) |
| | ERA5-S | 7 | 3072→96 | (81816, 11688, 23376) | Climate (3 Hours) |
| Multivariate Forecasting | ETTh1, ETTh2 | 7 | {96, 672}→{96, 192, 336, 720} | (8545, 2881, 2881) | Electricity (Hourly) |
| | ETTm1, ETTm2 | 7 | {96, 672}→{96, 192, 336, 720} | (34465, 11521, 11521) | Electricity (15 mins) |
| | ECL | 321 | {96, 672}→{96, 192, 336, 720} | (18317, 2633, 5261) | Electricity (Hourly) |
| | Traffic | 862 | {96, 672}→{96, 192, 336, 720} | (12185, 1757, 3509) | Transportation (Hourly) |
| | Weather | 21 | {96, 672}→{96, 192, 336, 720} | (36792, 5271, 10540) | Climate (10 mins) |
| | Solar-Energy | 137 | {96, 672}→{96, 192, 336, 720} | (36601, 5161, 10417) | Energy (10 mins) |
| | ERA5-MS | 7 | 3072→96 | (81816, 11688, 23376) | Climate (3 Hours) |
| | GTWSF | 3850 | 48→24 | (12280, 1755, 3509) | Wu et al. (2023) |
| Forecasting with Covariates | NP | 1+2 | 168→24 | (36500, 5219, 10460) | Electricity (Hourly) |
| | PJM | 1+2 | 168→24 | (36500, 5219, 10460) | Electricity (Hourly) |
| | BE | 1+2 | 168→24 | (36500, 5219, 10460) | Electricity (Hourly) |
| | FR | 1+2 | 168→24 | (36500, 5219, 10460) | Electricity (Hourly) |
| | DE | 1+2 | 168→24 | (36500, 5219, 10460) | Electricity (Hourly) |
| Pre-training | ERA5-Large | 4920 | 3072→96 | (81816, 11688, 23376) | Climate (3 Hours) |
| | UTSD | - | 2880→96 | (868778970, 96530996, -) | Liu et al. (2024c) |
| | LOTSA | - | 2880→96 | (231082956489, -, -) | Woo et al. (2024) |

Table 10: Performance robustness of Timer-XL. The prediction settings and results keep the same with Table 12. The standard deviation is obtained from three random seeds.

| Dataset | ECL | | ETTh1 | | Traffic | |
|---|---|---|---|---|---|---|
| Horizon | MSE | MAE | MSE | MAE | MSE | MAE |
| 96 | 0.127±0.001 | 0.219±0.001 | 0.364±0.002 | 0.397±0.001 | 0.340±0.002 | 0.238±0.001 |
| 192 | 0.145±0.001 | 0.236±0.001 | 0.405±0.002 | 0.424±0.001 | 0.360±0.001 | 0.247±0.001 |
| 336 | 0.159±0.001 | 0.252±0.001 | 0.427±0.003 | 0.439±0.002 | 0.377±0.002 | 0.256±0.002 |
| 720 | 0.187±0.003 | 0.277±0.003 | 0.439±0.002 | 0.459±0.004 | 0.418±0.003 | 0.279±0.002 |

| Dataset | Solar-Energy | | Weather | | ERA5-MS | |
|---|---|---|---|---|---|---|
| Horizon | MSE | MAE | MSE | MAE | MSE | MAE |
| 96 | 0.162±0.003 | 0.221±0.002 | 0.157±0.002 | 0.205±0.001 | 0.164±0.001 | 0.307±0.000 |
| 192 | 0.187±0.003 | 0.239±0.002 | 0.206±0.003 | 0.250±0.002 | | |
| 336 | 0.205±0.003 | 0.255±0.002 | 0.259±0.003 | 0.291±0.003 | | |
| 720 | 0.238±0.003 | 0.279±0.003 | 0.337±0.002 | 0.344±0.002 | | |

We adopt channel independence from Nie et al. (2022) in univariate time series forecasting. Based on the prevalence of patch-level tokenization in the time series field, we reproduce typical Transformers: PatchTST (2022), Timer (2024c), and UniTST (2024a) based on their official repositories, and keep their model hyperparameters and training configurations the same to evaluate the inherent capability of base models. The results of other baselines are based on the benchmark provided by Liu et al. (2023; 2024b); Ding et al. (2024); Wang et al. (2024b), which is fairly built on the configurations provided by their original paper. Detailed experimental configurations are provided in Table 11. We also report the standard deviations under three runs with different random seeds in Table 10, which exhibits that the performance of Timer-XL is stable.

For the metrics, we adopt the symmetric mean absolute percentage error (SMAPE), a metric that is independent of the numerical range, to evaluate one-for-all generalization performance on ERA5-Large. For other experiments, we adopt the root mean square error (MSE) and mean absolute error (MAE) that follows previous work. These metrics can be calculated as follows:

$$\text{SMAPE} = \frac{200}{T} \sum_{i=1}^{T} \frac{|\mathbf{X}_i - \widehat{\mathbf{X}}_i|}{|\mathbf{X}_i| + |\widehat{\mathbf{X}}_i|}, \ \text{MSE} = \sum_{i=1}^{T} |\mathbf{X}_i - \widehat{\mathbf{X}}_i|^2, \ \text{MAE} = \sum_{i=1}^{T} |\mathbf{X}_i - \widehat{\mathbf{X}}_i|.$$

Here $\mathbf{X} \in \mathbb{R}^T$ is a univariate time series and $\widehat{\mathbf{X}}$ is the corresponding prediction. For multivariate time series, we further calculate the mean metric in the variable dimension.

Table 11: Experimental configurations of Timer-XL and other baseline Transformers. All the experiments adopt the ADAM (2014) optimizer with the default hyperparameter $(\beta_1, \beta_2) = (0.9, 0.999)$.

| Experiment | Model | Dataset | Configuration | | | | | Training Process | | | |
|---|---|---|---|---|---|---|---|---|---|---|---|
| | | | L | D | $d_k$ | H | P | LR | Loss | Batch Size | Epochs |
| Univariate Forecasting | Timer-XL PatchTST | ECL | 3 | 512 | 64 | 8 | 96 | 0.0005 | MSE | 2048 | 10 |
| | | Traffic | 3 | 512 | 64 | 8 | 96 | 0.001 | MSE | 2048 | 10 |
| | | ETTh1 | 1 | 512 | 64 | 8 | 96 | 0.0005 | MSE | 256 | 10 |
| | | PEMS03 | 3 | 512 | 64 | 8 | 96 | 0.0005 | MSE | 2048 | 10 |
| | | ERA5-S | 1 | 512 | 64 | 8 | 96 | 0.0005 | MSE | 2048 | 10 |
| Multivariate Forecasting | Timer-XL UniTST Timer PatchTST | Global Temp. | 3 | 1024 | 128 | 8 | 24 | 0.0001 | MSE | 8 | 10 |
| | | Global Wind | 3 | 1024 | 128 | 8 | 24 | 0.0001 | MSE | 8 | 10 |
| | | ECL | 5 | 512 | 64 | 8 | 96 | 0.0005 | MSE | 4 | 10 |
| | | Traffic | 4 | 512 | 64 | 8 | 96 | 0.0005 | MSE | 4 | 10 |
| | | ETTh1 | 1 | 1024 | 128 | 8 | 96 | 0.0001 | MSE | 32 | 10 |
| | | Weather | 4 | 512 | 64 | 8 | 96 | 0.0005 | MSE | 32 | 10 |
| | | Solar. | 6 | 512 | 64 | 8 | 96 | 0.0001 | MSE | 16 | 10 |
| | | ERA5-MS | 3 | 512 | 64 | 8 | 96 | 0.0001 | MSE | 256 | 10 |
| Forecasting with Covariates | Timer-XL TimeXer Timer PatchTST | NP | 3 | 512 | 64 | 8 | 24 | 0.0001 | MSE | 4 | 10 |
| | | PJM | 2 | 512 | 64 | 8 | 24 | 0.0001 | MSE | 16 | 10 |
| | | BE | 2 | 512 | 64 | 8 | 24 | 0.0001 | MSE | 16 | 10 |
| | | FR | 2 | 512 | 64 | 8 | 24 | 0.0001 | MSE | 16 | 10 |
| | | DE | 2 | 512 | 64 | 8 | 24 | 0.0001 | MSE | 16 | 10 |
| Pre-training | Timer-XL | ERA5-Large | 4 | 512 | 64 | 8 | 96 | 0.0001 | MSE | 40960 | 10 |
| | PatchTST | | 4 | 512 | 64 | 8 | 96 | 0.0001 | MSE | 40960 | 10 |
| | Timer-XL | UTSD | 8 | 1024 | 128 | 8 | 96 | 0.00005 | MSE | 16384 | 10 |
| | Timer | (Liu et al., 2024c) | 8 | 1024 | 128 | 8 | 96 | 0.00005 | MSE | 16384 | 10 |
| | Timer-XL | | 8 | 1024 | 128 | 8 | 96 | 0.001 | MSE | 32768 | - |
| | Moirai$_{\text{Small}}$ | LOTSA | 6 | 384 | 64 | 6 | - | | | | |
| | Moirai$_{\text{Base}}$ | (Woo et al., 2024) | 12 | 768 | 64 | 12 | - | | Woo et al. (2024) | | |
| | Moirai$_{\text{Large}}$ | | 24 | 1024 | 64 | 16 | - | | | | |

$*$ $L$ is the layer number of Transformers, $D$ is the dimension of token embedding (the hidden dimension of FFN is set as $4D$), $d_k$ is the dimension of query, key, and value, $H$ is the multi-head number, $P$ is the patch size, and LR is the initial learning rate.

## C HYPERPARAMETER SENSITIVITY

We evaluate the hyperparameter sensitivity of Timer-XL on the ERA5-MS benchmark, as illustrated in Figure 8, concerning the following factors: the number of layers $L$, the patch size $P$, and the lookback length during inference. Our findings indicate that performance of Timer-XL generally improves with increases with $L$, suggesting that Timer-XL is a scalable deep forecaster. Furthermore, our analysis of the influence of $P$ reveals that the optimal patch size is generally close to the predicted length, since it avoid multi-step error accumulations. Toward better long-term forecasting performance, it leaves a future improvement to adopt different patch sizes of input and output tokens. Finally, we investigate the impact of input length during inference. We discover that the optimal lookback length of during is not necessarily the length during training. Given that decoder-only Transformers can accommodate inference inputs shorter than those used during training, this finding is noteworthy and indicates the potential to improve the performance.

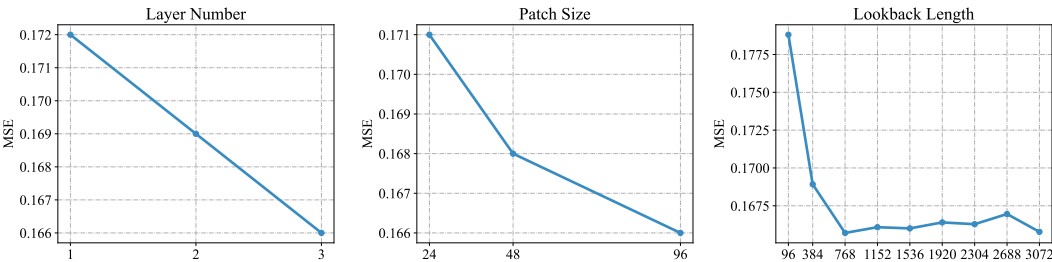

Figure 8: Hyperparameter sensitivity of Timer-XL (input-3072-pred-96 on ERA5-MS), including the number of Transformer blocks $L$, the patch size $P$, and the input lookback length during inference.

## D SHOWCASES

To facilitate a clear comparison among various models, we present additional prediction visualization from diverse datasets in Figure 9 and 10. Showcases are randomly selected from Timer-XL and the following time-series Transformers: PatchTST (2022), Timer (2024c), and UniTST (2024a). Among them, Timer-XL presents the most accurate predictions.

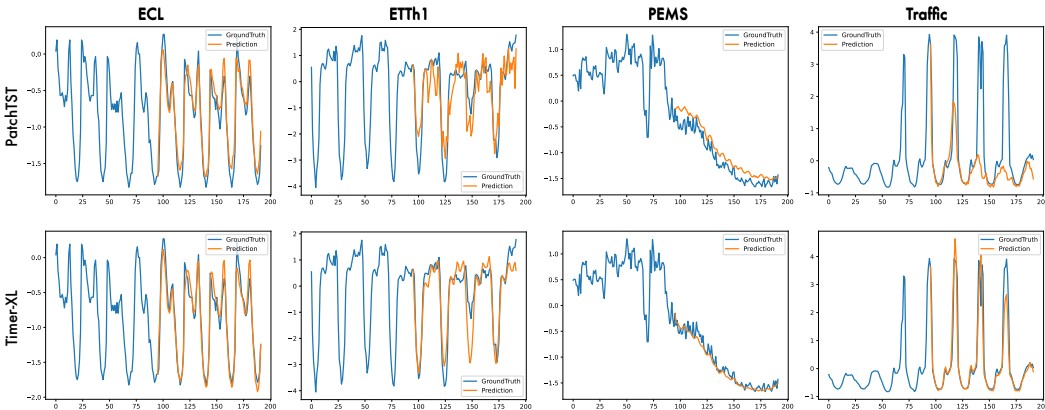

Figure 9: Visualization results on univariate time series dataset. We adopt the forecasting setting of 2880-pred-96 on ECL, ETTh1 and Traffic, and 2016-pred-96 on PEMS.

## E SUPPLEMENTARY RESULTS

### E.1 FULL RESULT OF MULTIVARIATE FORECASTING

Table 12 provides the complete results of the one-for-all multivariate forecasting benchmark across well-acknowledged datasets. We evaluate Timer-XL and baseline models by rolling forecasting: each

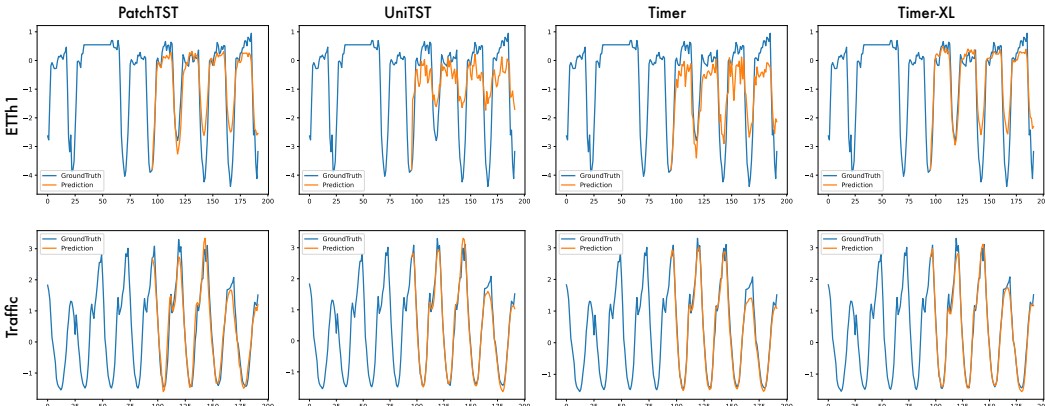

Figure 10: Visualization results on multivariate time series dataset. We adopt the forecasting setting of 672-pred-96 on ETTh1 (7 Variables) and Traffic (862 Variables).

model is trained with input length 672 and output length 96, and the predicted values are integrated as part of the input in the next iteration until reaching the desired forecast length in $\{96, 192, 336, 720\}$.

We highlight that this benchmark evaluates the fundamental model versatility of deep forecasters, which aims to break the awkward situation of extensive training and model storage in pursuit of better practice for real-world forecasting requirements. On this benchmark, time-series Transformers significantly stand out from other baseline models, and our proposed Timer-XL can achieve state-of-the-art performance, making it a nice fundamental backbone of a one-for-all forecaster.

### E.2 Full Result of Zero-Shot Forecasting

Table 13 provides the full results of zero-shot forecasting on the benchmark from Wu et al. (2022). We build Timer-XL based on the configuration in Table 11, which is pre-trained on the aggregated datasets of UTSD (Liu et al., 2024c) and LOTSA (Woo et al., 2024). The patch size of Timer-XL is set as 96 and we conduct rolling forecast to obtain the desired forecast length in $\{96, 192, 336, 720\}$.

We evaluate most advanced large models based on their official model checkpoints, including Time-MoE (Shi et al., 2024), Moirai (Woo et al., 2024), TimesFM (Das et al., 2023), MOMENT Goswami et al. (2024), and Chronos (Ansari et al., 2024). We conduct zero-shot evaluations on datasets that are not included during the pre-training of corresponding models. For each of the evaluated model, we use their maximum input length during inference. The metric (MSE/MAE) is averaged from all predicted windows in the test split.

### E.3 Ablation Study of TimeAttention

We conduct evaluations on TimeAttention to validate the effectiveness of position embeddings. As for variable embedding, the distinction between endogenous and exogenous variables can improve performance. Based on our observation of the learned $u > v$, we find that the token reasonably pays more attention to tokens of the endogenous variable. It leaves a prior to mask out minor dependencies that focuses less on exogenous variables. For the temporal dimension, other position embeddings are inferior to RoPE, since it uses the affine transformation, while others are additive, and thereby less confused with the same additive embedding for variables.

### E.4 Supplementary Results of Long-Context Forecasting

Long context is a basic indicator of foundation models, which can support emergence capabilities such as prompting, in-context learning, retrieval-augmented generation, etc. However, the long-context forecasting paradigm receives less attention in the current community, which can be due to the lack of benchmarks. In the meteorological ERA5, it is necessary to support the context of more than years to contain a specific cycle (such as El Nino). In Table 15, the performance of Timer-XL and DLinear generally improves with the increased context length. By contrast, it reveals the performance

Table 12: Full multivariate forecasting results: we conduct rolling forecast with a single model trained on each dataset (lookback length is 672) and accomplish four forecast lengths in $\{96, 192, 336, 720\}$.

| Models | | Timer-XL (Ours) | | Timer (2024c) | | UniTST (2024a) | | iTransformer (2023) | | DLinear (2023) | | PatchTST (2022) | | TimesNet (2022) | | Stationary (2022b) | | Autoformer (2021) | |
|---|---|---|---|---|---|---|---|---|---|---|---|---|---|---|---|---|---|---|---|
| Metric | | MSE | MAE | MSE | MAE | MSE | MAE | MSE | MAE | MSE | MAE | MSE | MAE | MSE | MAE | MSE | MAE | MSE | MAE |
| ETTh1 | 96 | **0.364** | **0.397** | 0.371 | 0.404 | 0.379 | 0.415 | 0.387 | 0.418 | 0.369 | 0.400 | 0.373 | 0.403 | 0.452 | 0.463 | 0.452 | 0.478 | 0.467 | 0.499 |
| | 192 | **0.405** | 0.424 | 0.407 | 0.429 | 0.415 | 0.438 | 0.416 | 0.437 | **0.405** | **0.422** | 0.405 | 0.425 | 0.474 | 0.477 | 0.484 | 0.510 | 0.492 | 0.523 |
| | 336 | 0.427 | **0.439** | 0.434 | 0.445 | 0.440 | 0.454 | 0.434 | 0.450 | 0.435 | 0.445 | 0.423 | 0.440 | 0.493 | 0.489 | 0.511 | 0.522 | 0.519 | 0.531 |
| | 720 | **0.439** | **0.459** | 0.461 | 0.466 | 0.482 | 0.482 | 0.447 | 0.473 | 0.493 | 0.508 | 0.445 | 0.471 | 0.560 | 0.534 | 0.571 | 0.543 | 0.589 | 0.560 |
| | Avg | **0.409** | **0.430** | 0.418 | 0.436 | 0.429 | 0.447 | 0.421 | 0.445 | 0.426 | 0.444 | 0.412 | 0.435 | 0.495 | 0.491 | 0.505 | 0.513 | 0.517 | 0.528 |
| ETTh2 | 96 | **0.277** | **0.343** | 0.285 | 0.344 | 0.343 | 0.398 | 0.304 | 0.362 | 0.305 | 0.371 | 0.289 | 0.347 | 0.340 | 0.374 | 0.348 | 0.403 | 0.358 | 0.397 |
| | 192 | **0.348** | **0.391** | 0.365 | 0.400 | 0.376 | 0.420 | 0.372 | 0.407 | 0.412 | 0.439 | 0.360 | 0.393 | 0.402 | 0.414 | 0.408 | 0.448 | 0.435 | 0.451 |
| | 336 | **0.375** | **0.418** | 0.412 | 0.440 | 0.399 | 0.435 | 0.418 | 0.440 | 0.527 | 0.508 | 0.389 | 0.420 | 0.452 | 0.452 | 0.424 | 0.457 | 0.454 | 0.475 |
| | 720 | 0.409 | 0.458 | 0.468 | 0.487 | 0.419 | 0.457 | 0.463 | 0.476 | 0.830 | 0.653 | **0.398** | **0.440** | 0.462 | 0.468 | 0.448 | 0.476 | 0.479 | 0.492 |
| | Avg | **0.352** | 0.402 | 0.382 | 0.418 | 0.384 | 0.428 | 0.389 | 0.421 | 0.518 | 0.493 | 0.359 | **0.400** | 0.414 | 0.427 | 0.407 | 0.446 | 0.431 | 0.454 |
| ETTm1 | 96 | 0.290 | 0.341 | **0.281** | **0.338** | 0.289 | 0.348 | 0.311 | 0.365 | 0.307 | 0.350 | 0.285 | 0.346 | 0.338 | 0.375 | 0.414 | 0.414 | 0.466 | 0.466 |
| | 192 | 0.337 | 0.369 | 0.330 | **0.368** | 0.332 | 0.375 | 0.353 | 0.390 | 0.337 | **0.368** | **0.329** | 0.372 | 0.371 | 0.387 | 0.524 | 0.482 | 0.504 | 0.496 |
| | 336 | 0.374 | 0.392 | 0.367 | 0.393 | 0.365 | 0.397 | 0.387 | 0.411 | 0.366 | **0.387** | **0.363** | 0.394 | 0.410 | 0.411 | 0.541 | 0.497 | 0.574 | 0.530 |
| | 720 | 0.437 | 0.428 | 0.432 | 0.433 | 0.421 | 0.431 | 0.452 | 0.445 | **0.419** | **0.419** | 0.421 | 0.426 | 0.478 | 0.450 | 0.578 | 0.509 | 0.596 | 0.558 |
| | Avg | 0.359 | 0.382 | 0.352 | 0.383 | 0.352 | 0.388 | 0.376 | 0.403 | 0.357 | **0.381** | **0.349** | 0.385 | 0.399 | 0.406 | 0.514 | 0.475 | 0.535 | 0.512 |
| ETTm2 | 96 | **0.175** | **0.257** | 0.175 | **0.257** | 0.171 | 0.260 | 0.183 | 0.272 | **0.167** | 0.263 | 0.172 | 0.259 | 0.187 | 0.267 | 0.237 | 0.306 | 0.255 | 0.339 |
| | 192 | 0.242 | 0.301 | 0.239 | 0.301 | **0.228** | **0.230** | 0.250 | 0.315 | 0.230 | 0.311 | 0.233 | 0.299 | 0.249 | 0.309 | 0.330 | 0.387 | 0.279 | 0.335 |
| | 336 | 0.293 | 0.337 | 0.293 | 0.342 | 0.282 | 0.336 | 0.311 | 0.356 | 0.298 | 0.361 | **0.280** | **0.331** | 0.321 | 0.351 | 0.404 | 0.424 | 0.331 | 0.374 |
| | 720 | 0.376 | 0.390 | 0.392 | 0.407 | 0.380 | 0.398 | 0.417 | 0.419 | 0.432 | 0.446 | **0.357** | **0.382** | 0.497 | 0.403 | 0.525 | 0.486 | 0.413 | 0.450 |
| | Avg | 0.271 | 0.322 | 0.275 | 0.327 | 0.265 | **0.306** | 0.290 | 0.340 | 0.282 | 0.345 | **0.261** | 0.318 | 0.314 | 0.333 | 0.374 | 0.401 | 0.320 | 0.374 |
| ECL | 96 | **0.127** | **0.219** | 0.129 | 0.221 | 0.130 | 0.225 | 0.133 | 0.229 | 0.138 | 0.238 | 0.132 | 0.232 | 0.184 | 0.288 | 0.185 | 0.287 | 0.256 | 0.357 |
| | 192 | **0.145** | **0.236** | 0.148 | 0.239 | 0.150 | 0.244 | 0.158 | 0.258 | 0.152 | 0.251 | 0.151 | 0.250 | 0.192 | 0.295 | 0.282 | 0.368 | 0.291 | 0.376 |
| | 336 | **0.159** | **0.252** | 0.164 | 0.256 | 0.166 | 0.262 | 0.168 | 0.262 | 0.167 | 0.268 | 0.171 | 0.272 | 0.200 | 0.303 | 0.289 | 0.377 | 0.290 | 0.379 |
| | 720 | **0.187** | **0.277** | 0.201 | 0.289 | 0.206 | 0.297 | 0.205 | 0.294 | 0.203 | 0.302 | 0.222 | 0.318 | 0.228 | 0.325 | 0.305 | 0.399 | 0.320 | 0.403 |
| | Avg | **0.155** | **0.246** | 0.161 | 0.251 | 0.163 | 0.257 | 0.164 | 0.258 | 0.165 | 0.265 | 0.169 | 0.268 | 0.201 | 0.303 | 0.265 | 0.358 | 0.289 | 0.379 |
| Traffic | 96 | **0.340** | **0.238** | 0.348 | 0.240 | 0.359 | 0.250 | 0.353 | 0.259 | 0.399 | 0.285 | 0.359 | 0.255 | 0.593 | 0.315 | 0.610 | 0.322 | 0.675 | 0.412 |
| | 192 | **0.360** | **0.247** | 0.369 | 0.250 | 0.373 | 0.257 | 0.373 | 0.267 | 0.409 | 0.290 | 0.377 | 0.265 | 0.596 | 0.317 | 0.626 | 0.346 | 0.679 | 0.423 |
| | 336 | **0.377** | **0.256** | 0.388 | 0.260 | 0.386 | 0.265 | 0.386 | 0.275 | 0.422 | 0.297 | 0.393 | 0.276 | 0.600 | 0.319 | 0.633 | 0.352 | 0.688 | 0.440 |
| | 720 | **0.418** | **0.279** | 0.431 | 0.285 | 0.421 | 0.286 | 0.425 | 0.296 | 0.461 | 0.319 | 0.436 | 0.305 | 0.619 | 0.335 | 0.651 | 0.366 | 0.693 | 0.457 |
| | Avg | **0.374** | **0.255** | 0.384 | 0.259 | 0.385 | 0.265 | 0.384 | 0.274 | 0.423 | 0.298 | 0.391 | 0.275 | 0.602 | 0.322 | 0.630 | 0.347 | 0.684 | 0.433 |
| Weather | 96 | 0.157 | 0.205 | 0.151 | **0.202** | 0.152 | 0.206 | 0.174 | 0.225 | 0.169 | 0.229 | **0.149** | **0.202** | 0.169 | 0.228 | 0.185 | 0.241 | 0.355 | 0.409 |
| | 192 | 0.206 | 0.250 | 0.196 | **0.245** | 0.198 | 0.249 | 0.227 | 0.268 | 0.211 | 0.268 | **0.194** | **0.245** | 0.222 | 0.269 | 0.286 | 0.325 | 0.421 | 0.450 |
| | 336 | 0.259 | 0.291 | 0.249 | 0.288 | 0.251 | 0.291 | 0.290 | 0.309 | 0.258 | 0.306 | **0.244** | **0.285** | 0.290 | 0.310 | 0.323 | 0.347 | 0.452 | 0.465 |
| | 720 | 0.337 | 0.344 | 0.330 | 0.344 | 0.322 | 0.340 | 0.374 | 0.360 | 0.320 | 0.362 | **0.317** | **0.338** | 0.376 | 0.364 | 0.436 | 0.401 | 0.513 | 0.496 |
| | Avg | 0.240 | 0.273 | 0.232 | 0.270 | 0.231 | 0.272 | 0.266 | 0.291 | 0.239 | 0.291 | **0.226** | **0.268** | 0.264 | 0.293 | 0.308 | 0.329 | 0.435 | 0.455 |
| Solar-Energy | 96 | **0.162** | **0.221** | 0.212 | 0.230 | 0.190 | 0.240 | 0.183 | 0.265 | 0.193 | 0.258 | 0.168 | 0.237 | 0.180 | 0.272 | 0.199 | 0.290 | 0.206 | 0.296 |
| | 192 | **0.187** | **0.239** | 0.232 | 0.246 | 0.223 | 0.264 | 0.205 | 0.283 | 0.214 | 0.274 | 0.189 | 0.257 | 0.199 | 0.286 | 0.243 | 0.307 | 0.254 | 0.328 |
| | 336 | **0.205** | 0.255 | 0.237 | 0.253 | 0.250 | 0.283 | 0.224 | 0.299 | 0.233 | 0.291 | 0.212 | 0.277 | 0.220 | 0.301 | 0.264 | 0.322 | 0.272 | 0.330 |
| | 720 | **0.238** | 0.279 | 0.252 | **0.266** | 0.292 | 0.311 | 0.239 | 0.316 | 0.246 | 0.307 | 0.240 | 0.305 | 0.251 | 0.321 | 0.310 | 0.339 | 0.326 | 0.347 |
| | Avg | **0.198** | **0.249** | 0.233 | **0.249** | 0.241 | 0.275 | 0.213 | 0.291 | 0.222 | 0.283 | 0.202 | 0.269 | 0.213 | 0.295 | 0.254 | 0.315 | 0.265 | 0.325 |
| 1st Count | | **23** | **21** | 1 | 8 | 1 | 2 | 0 | 0 | 3 | 5 | 14 | 9 | 0 | 0 | 0 | 0 | 0 | 0 |

Table 13: Full results of zero-shot forecasting. A lower MSE or MAE indicates a better prediction. 1st Count represents the number of wins achieved by a model under all prediction lengths and datasets.

| Models | | Timer-XL_Base (Ours) | | Time-MoE_Base (2024) | | Time-MoE_Large (2024) | | Time-MoE_Ultra (2024) | | Moirai_Small (2024) | | Moirai_Base (2024) | | Moirai_Large (2024) | | TimesFM (2023) | | MOMENT (2024) | | Chronos_Base (2024) | | Chronos_Large (2024) | |
|---|---|---|---|---|---|---|---|---|---|---|---|---|---|---|---|---|---|---|---|---|---|---|---|
| Metric | | MSE | MAE | MSE | MAE | MSE | MAE | MSE | MAE | MSE | MAE | MSE | MAE | MSE | MAE | MSE | MAE | MSE | MAE | MSE | MAE | MSE | MAE |
| ETTm1 | 96 | 0.317 | 0.356 | 0.338 | 0.368 | 0.309 | 0.357 | **0.281** | **0.341** | 0.418 | 0.392 | 0.363 | 0.356 | 0.380 | 0.361 | 0.361 | 0.370 | 0.654 | 0.527 | 0.454 | 0.408 | 0.457 | 0.403 |
| | 192 | 0.358 | 0.381 | 0.353 | 0.388 | 0.346 | 0.381 | **0.305** | **0.358** | 0.431 | 0.405 | 0.388 | 0.375 | 0.412 | 0.383 | 0.414 | 0.405 | 0.662 | 0.532 | 0.567 | 0.477 | 0.530 | 0.450 |
| | 336 | 0.386 | 0.401 | 0.381 | 0.413 | 0.373 | 0.408 | **0.369** | **0.395** | 0.433 | 0.412 | 0.416 | **0.392** | 0.436 | 0.400 | 0.445 | 0.429 | 0.672 | 0.537 | 0.662 | 0.525 | 0.577 | 0.481 |
| | 720 | **0.430** | 0.431 | 0.504 | 0.493 | 0.475 | 0.477 | 0.469 | 0.472 | 0.462 | 0.432 | 0.460 | 0.418 | 0.462 | 0.420 | 0.512 | 0.471 | 0.692 | 0.551 | 0.900 | 0.591 | 0.660 | 0.526 |
| | Avg | 0.373 | 0.392 | 0.394 | 0.415 | 0.376 | 0.405 | **0.356** | 0.391 | 0.436 | 0.410 | 0.406 | **0.385** | 0.422 | 0.391 | 0.433 | 0.418 | 0.670 | 0.536 | 0.645 | 0.500 | 0.555 | 0.465 |
| ETTm2 | 96 | **0.189** | 0.277 | 0.201 | 0.291 | 0.197 | 0.286 | 0.198 | 0.288 | 0.214 | 0.288 | 0.205 | 0.273 | 0.211 | 0.274 | 0.202 | **0.270** | 0.260 | 0.335 | 0.199 | 0.274 | 0.197 | 0.271 |
| | 192 | 0.241 | 0.315 | 0.258 | 0.334 | 0.250 | 0.322 | **0.235** | **0.312** | 0.284 | 0.332 | 0.275 | 0.316 | 0.281 | 0.318 | 0.289 | 0.321 | 0.289 | 0.350 | 0.261 | 0.322 | 0.254 | 0.314 |
| | 336 | **0.286** | **0.348** | 0.324 | 0.373 | 0.337 | 0.375 | 0.293 | 0.348 | 0.331 | 0.362 | 0.329 | 0.350 | 0.341 | 0.355 | 0.360 | 0.366 | 0.324 | 0.369 | 0.326 | 0.366 | 0.313 | 0.353 |
| | 720 | **0.375** | **0.402** | 0.488 | 0.464 | 0.480 | 0.461 | 0.427 | 0.428 | 0.402 | 0.408 | 0.437 | 0.411 | 0.485 | 0.428 | 0.462 | 0.430 | 0.394 | 0.409 | 0.455 | 0.439 | 0.416 | 0.415 |
| | Avg | **0.273** | **0.336** | 0.317 | 0.365 | 0.316 | 0.361 | 0.288 | 0.344 | 0.307 | 0.347 | 0.311 | 0.337 | 0.329 | 0.343 | 0.328 | 0.346 | 0.316 | 0.365 | 0.310 | 0.350 | 0.295 | 0.338 |
| ETTh1 | 96 | 0.369 | 0.391 | 0.357 | 0.381 | 0.350 | 0.382 | **0.349** | **0.379** | 0.401 | 0.402 | 0.376 | 0.392 | 0.381 | 0.388 | 0.414 | 0.404 | 0.688 | 0.557 | 0.440 | 0.393 | 0.441 | 0.390 |
| | 192 | 0.405 | 0.413 | **0.384** | **0.404** | 0.388 | 0.412 | 0.395 | 0.413 | 0.435 | 0.421 | 0.412 | 0.413 | 0.434 | 0.415 | 0.465 | 0.434 | 0.688 | 0.560 | 0.492 | 0.426 | 0.502 | 0.524 |
| | 336 | 0.418 | **0.423** | 0.411 | 0.434 | 0.411 | 0.430 | 0.447 | 0.453 | 0.438 | 0.434 | 0.433 | 0.428 | 0.485 | 0.445 | 0.503 | 0.456 | 0.675 | 0.563 | 0.550 | 0.462 | 0.576 | 0.467 |
| | 720 | **0.423** | **0.441** | 0.449 | 0.477 | 0.427 | 0.455 | 0.457 | 0.462 | 0.439 | 0.454 | 0.447 | 0.444 | 0.611 | 0.510 | 0.511 | 0.481 | 0.683 | 0.585 | 0.882 | 0.591 | 0.835 | 0.583 |
| | Avg | 0.404 | **0.417** | 0.400 | 0.424 | 0.394 | 0.419 | 0.412 | 0.426 | 0.428 | 0.427 | 0.417 | 0.419 | 0.480 | 0.439 | 0.473 | 0.443 | 0.683 | 0.566 | 0.591 | 0.468 | 0.588 | 0.466 |
| ETTh2 | 96 | **0.283** | 0.342 | 0.305 | 0.359 | 0.302 | 0.354 | 0.292 | 0.352 | 0.297 | 0.336 | 0.294 | **0.330** | 0.296 | **0.330** | 0.315 | 0.349 | 0.342 | 0.396 | 0.308 | 0.343 | 0.320 | 0.345 |
| | 192 | **0.340** | 0.379 | 0.351 | 0.386 | 0.364 | 0.385 | 0.347 | 0.379 | 0.368 | 0.381 | 0.365 | 0.375 | 0.361 | **0.371** | 0.388 | 0.395 | 0.354 | 0.402 | 0.384 | 0.392 | 0.406 | 0.399 |
| | 336 | 0.366 | 0.400 | 0.391 | 0.418 | 0.417 | 0.425 | 0.406 | 0.419 | 0.370 | 0.393 | 0.376 | 0.390 | 0.390 | **0.390** | 0.422 | 0.427 | **0.356** | 0.407 | 0.429 | 0.430 | 0.492 | 0.453 |
| | 720 | 0.397 | 0.431 | 0.419 | 0.454 | 0.537 | 0.496 | 0.439 | 0.447 | 0.411 | **0.426** | 0.416 | 0.433 | 0.423 | **0.418** | 0.443 | 0.454 | 0.395 | 0.434 | 0.501 | 0.477 | 0.603 | 0.511 |
| | Avg | 0.347 | 0.388 | 0.366 | 0.404 | 0.405 | 0.415 | 0.371 | 0.399 | 0.361 | 0.384 | 0.362 | **0.382** | 0.367 | **0.377** | 0.392 | 0.406 | 0.361 | 0.409 | 0.405 | 0.410 | 0.455 | 0.427 |
| ECL | 96 | **0.141** | 0.237 | - | - | - | - | - | - | 0.189 | 0.280 | 0.160 | 0.250 | 0.153 | 0.241 | - | - | 0.745 | 0.680 | 0.154 | 0.231 | 0.152 | **0.229** |
| | 192 | **0.159** | 0.254 | - | - | - | - | - | - | 0.205 | 0.292 | 0.175 | 0.263 | 0.169 | 0.255 | - | - | 0.755 | 0.683 | 0.179 | 0.254 | 0.172 | **0.250** |
| | 336 | **0.177** | **0.272** | - | - | - | - | - | - | 0.221 | 0.307 | 0.187 | 0.277 | 0.187 | 0.273 | - | - | 0.766 | 0.687 | 0.214 | 0.284 | 0.203 | 0.276 |
| | 720 | **0.219** | **0.308** | - | - | - | - | - | - | 0.258 | 0.335 | 0.228 | 0.309 | 0.237 | 0.313 | - | - | 0.794 | 0.696 | 0.311 | 0.346 | 0.289 | 0.337 |
| | Avg | **0.174** | 0.278 | - | - | - | - | - | - | 0.218 | 0.303 | 0.187 | 0.274 | 0.186 | **0.270** | - | - | 0.765 | 0.686 | 0.214 | 0.278 | 0.204 | 0.273 |
| Weather | 96 | 0.171 | 0.225 | 0.160 | 0.214 | 0.159 | 0.213 | **0.157** | **0.211** | 0.198 | 0.222 | 0.220 | 0.217 | 0.199 | **0.211** | - | - | 0.243 | 0.255 | 0.203 | 0.238 | 0.194 | 0.235 |
| | 192 | 0.221 | 0.271 | 0.210 | 0.260 | 0.215 | 0.266 | **0.208** | **0.256** | 0.247 | 0.265 | 0.271 | 0.259 | 0.246 | **0.251** | - | - | 0.278 | 0.329 | 0.256 | 0.290 | 0.249 | 0.285 |
| | 336 | 0.274 | **0.311** | 0.274 | 0.309 | 0.291 | 0.322 | **0.255** | **0.290** | 0.283 | 0.303 | 0.286 | 0.297 | 0.274 | 0.291 | - | - | 0.306 | 0.346 | 0.314 | 0.336 | 0.302 | 0.327 |
| | 720 | 0.356 | 0.370 | 0.418 | 0.405 | 0.415 | 0.400 | 0.405 | 0.397 | 0.373 | 0.354 | 0.373 | 0.354 | **0.337** | **0.340** | - | - | 0.350 | 0.374 | 0.397 | 0.396 | 0.372 | 0.378 |
| | Avg | **0.256** | 0.294 | 0.265 | 0.297 | 0.270 | 0.300 | **0.256** | 0.288 | 0.275 | 0.286 | 0.287 | **0.281** | 0.264 | **0.273** | - | - | 0.294 | 0.326 | 0.292 | 0.315 | 0.279 | 0.306 |
| 1st Count | | **15** | **10** | 2 | 1 | 3 | 0 | 10 | 7 | 0 | 0 | 0 | 5 | 1 | **10** | 0 | 1 | 2 | 0 | 0 | 0 | 0 | 2 |

∗ Dataset for pre-training is not evaluated on corresponding models, which is denoted by a dash (−).

∗ Traffic from (PEMS) is generally used during the pre-training of large models and thus not evaluated here.

∗ Our model checkpoint is available at https://huggingface.co/thuml/timer-base-84m.

Table 14: Embedding ablation in TimeAttention. For the temporal dimension, we compare prevalent relative and absolute position embeddings. As for the variable dimension, we explore the effectiveness of the variable embedding that distinguishes endogenous and exogenous variables.

| Design | Temporal | Variable | Traffic | | Weather | | Solar-Energy | | ERA5-MS | |
|---|---|---|---|---|---|---|---|---|---|---|
| | | | MSE | MAE | MSE | MAE | MSE | MAE | MSE | MAE |
| **Timer-XL** | **RoPE (2024)** | **with** | **0.340** | **0.238** | **0.157** | **0.205** | **0.162** | 0.221 | **0.164** | 0.307 |
| Replace | ALiBi (2021) | with | 0.351 | 0.246 | 0.162 | 0.212 | 0.188 | 0.210 | 0.167 | 0.308 |
| | Relative (2020) | with | 0.361 | 0.250 | 0.163 | 0.214 | 0.197 | 0.215 | 0.168 | 0.309 |
| | Absolute (2017) | with | 0.381 | 0.270 | 0.159 | 0.207 | 0.171 | **0.204** | 0.165 | **0.306** |
| w/o | RoPE (2024) | w/o | 0.361 | 0.254 | 0.171 | 0.217 | 0.181 | 0.221 | 0.235 | 0.373 |
| | w/o | w/o | 0.363 | 0.253 | 0.164 | 0.215 | 0.194 | 0.215 | 0.167 | 0.309 |

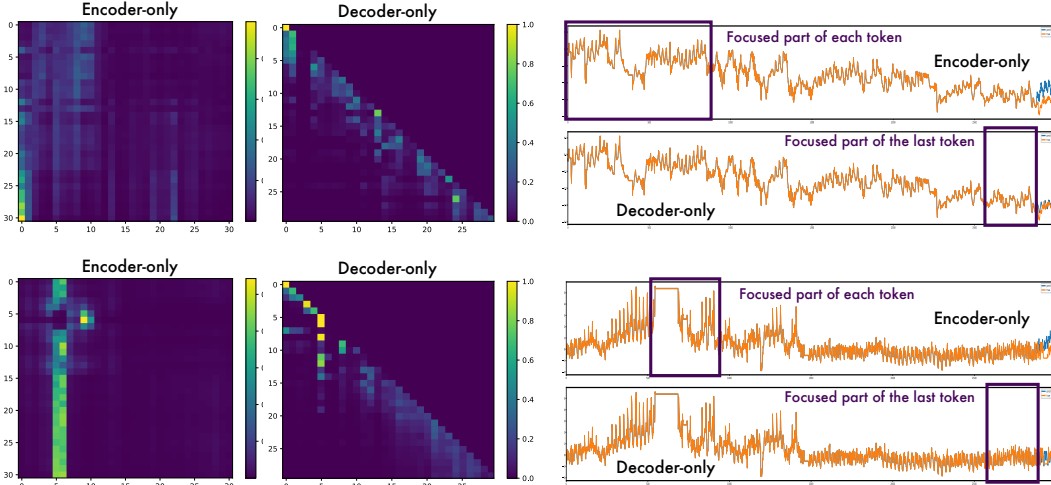

Figure 11: Case studies of learned attention in encoder-/decoder-only Transformers.

degradation of PatchTST. Similar to the observations in Figure 3, the encoder-only architecture produces inferior predictions after thousands of time points, which can be concealed due to the short context adopted in previous benchmarks. Although PatchTST has conducted an initial exploration in the context of hundreds of time points, it inappropriately works in ever-long contexts. Therefore, we believe that context bottlenecks deserve further exploration in this community.

Table 15: Performance on ERA5 (pred-1day). Lookback lengths vary from daily to yearly contexts.

| Models | Timer-XL | PatchTST | DLinear |
|---|---|---|---|
| Metric | MSE \| MAE | MSE \| MAE | MSE \| MAE |
| Lookback-8 (1 Day) | 0.0847 \| 0.2100 | 0.0897 \| 0.2196 | 0.0970 \| 0.2276 |
| Lookback-32 (4 Day) | 0.0713 \| 0.1928 | 0.0778 \| 0.2080 | 0.0841 \| 0.2113 |
| Lookback-56 (1 Week) | 0.0688 \| 0.1891 | 0.0785 \| 0.2082 | 0.0814 \| 0.2081 |
| Lookback-224 (1 Month) | 0.0675 \| 0.1868 | 0.0745 \| 0.2042 | 0.0788 \| 0.2048 |
| Lookback-960 (4 Month) | 0.0667 \| 0.1863 | 0.1194 \| 0.2696 | 0.0773 \| 0.2031 |
| Lookback-2944 (1 Year) | 0.0663 \| 0.1857 | 0.1109 \| 0.2638 | 0.0763 \| 0.2024 |

**Representation Analysis**   We further delve into long-context modeling from the perspective of learned representations. As shown in Figure 11, the decoder-only model can selectively focus on the previous context while PatchTST wrongly focuses on noisy parts. Since causality is the basis of forecasting, using causal masks leads to coherent token embeddings, while the unmasked attention mechanism may break the causality and prevent the model from telling each tokens.

**Normalization**   Section 4.1 has discussed instance normalization (Kim et al., 2021). It generally improves the performance of the previous encoder-only Transformers but leads to special problems in decoder-only Transformers (e.g., unmatched statistics in multi-step autoregression). However, it is indicative that Timer-XL without ReVIN can achieve competitive performance on well-acknowledged benchmarks in Table 16, while the performance of PatchTST may heavily rely on this normalization.

### E.5   ILLUSTRATION OF TIMEATTENTION

Although the formulation to generalize from 1D sequences to multivariate time series is straightforward, Timer-XL is built on a decoder-only Transformer, an underexploited backbone among current time series models. As shown in Figure 12, challenges lie in capturing fine-grained dependencies between all variables in the patch level, while maintaining temporal causality in multiple sequences.

Table 16: Evaluations (672-pred-96) on the effect of ReVIN (Kim et al., 2021) on Transformers.

| Models | Timer-XL with ReVIN | Timer-XL w/o ReVIN | PatchTST with ReVIN | PatchTST w/o ReVIN |
|---|---|---|---|---|
| Metric | MSE \| MAE | MSE \| MAE | MSE \| MAE | MSE \| MAE |
| ETTh1 | 0.364 \| 0.397 | 0.370 \| 0.401 | 0.370 \| 0.399 | 0.421 \| 0.448 |
| Weather | 0.157 \| 0.205 | 0.151 \| 0.205 | 0.149 \| 0.198 | 0.173 \| 0.242 |
| ECL | 0.127 \| 0.219 | 0.130 \| 0.225 | 0.129 \| 0.222 | 0.138 \| 0.244 |

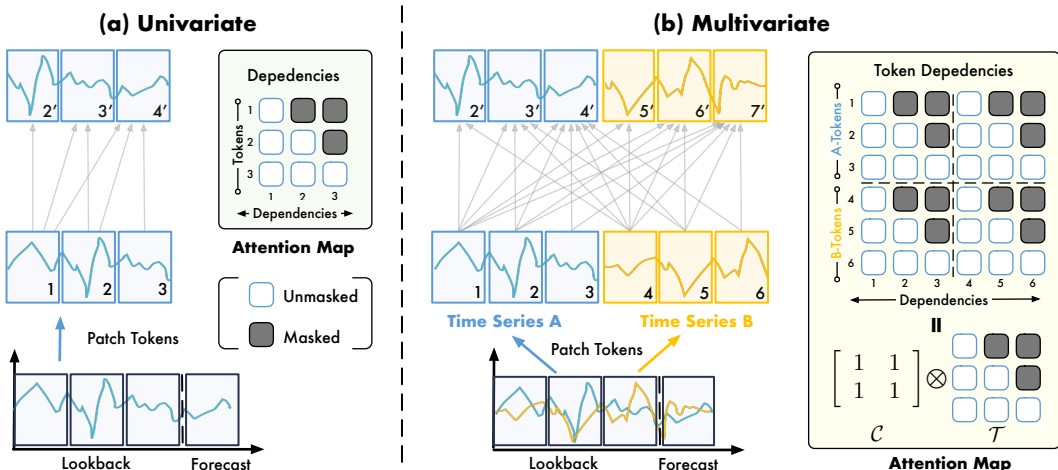

Figure 12: Illustration of TimeAttention for modeling univariate and multivariate time series.

Technically, we introduce the masking formulation, whose key lies in the grouped causality of flattened 2D sequences. We derive it based on the Kronecker Product, which disentangles the large attention map into formalizable temporal and variable dependencies. It can be naturally extended to covariates or pre-defined variable dependencies, which may inspire a lot of future explorations.

# F    LIMITATIONS

Timer-XL is a unified model for time series forecasting. It can be used for task-specific training or scalable pre-training, handling varying-length and multivariate time series. As an autoregressive model, Timer-XL necessitates iterative generation for long-term forecasting, which may lead to error accumulation and inflexibility in the output length. In the future, we plan to incorporate multi-resolution patches for input and output series. Furthermore, given that Timer-XL explicitly captures fine-grained token dependencies, there remains significant potential to reduce the complexity of TimeAttention, particularly in high-dimensional and lengthy time series. Finally, we will investigate the factors contributing to the stagnation of Transformer performance in extremely long contexts, and seek insights in the time series modality to improve context efficiency.

