# OpenReview forum: "Timer-XL: Long-Context Transformers for Unified Time Series Forecasting"
_ICLR.cc/2025/Conference — ICLR 2025 Poster_

### Official Review · Reviewer_V4mu · 2024-10-31

**Soundness:** 3
**Presentation:** 3
**Contribution:** 3
**Rating:** 6
**Confidence:** 4

**Summary:**

This paper introduces Timer-XL, a method for unified time series forecasting, which addresses challenges in hard to uniformly predict 1D and 2D time series. It utilizes generative Transformer with relative position embedding to capture the correlations between the temporal and variable dimensions. Also, it proposes a novel TimeAttention to capture causal patch-wise dependencies within and among all variables. Experiments demonstrate that Timer-XL achieves state-of-the-art performance across challenging forecasting benchmarks through a unified approach,

**Strengths:**

1. The manuscript demonstrates a high level of completeness.
2. It provides an effective large-scale framework for advancing large models in time series analysis.
3. The empirical evaluation is comprehensive and promising results are shown.

**Weaknesses:**

1. My biggest question lies with Equations 4, 5, 7, and 8. Since Timer-XL is a unified time series framework, these equations include processing steps that sort each time series. For example, with 𝑁 time series, does the order after flattening the sequences significantly affect the causal relationship in Equations 4 and 5? The same question applies to Equations 7 and 8.
2. As a unified time series framework, the author needs to compare it with Moirai in a zero-shot scenario. In Figure 5 of the manuscript, a corresponding comparison experiment should be added.

**Questions:**

Please refer to Weaknesses.

---

> ### Author Response · Authors · 2024-11-21
> **Response to Reviewer V4mu**
>
> Many thanks to Reviewer V4mu for providing a detailed review and recognizing our contributions.
>
> **Q1**: How does the order of the flattened sequence influence the equations?
>
> It is a very insightful question. Given $N$ time series, TimeAttention has the following properties:
>
> * **Avoid Permutation-Invariance on the Temporal Dimension**: Self-attention is permutation-invariant, which contradicts the causality in the temporal dimension.
> * **Permutation-Equivariance of Multiple Variables**: Formally, $\operatorname{Attn}(\mathcal{P}(V1,...,VN)) = \mathcal{P}(\operatorname{Attn}(V1, ..., VN))$, where $\mathcal{P}$ is any permutation of $N$ variables. With this property, the input and output of variables have a correspondence in position: Shuffling the input order of variables should be reflected in the output order of variables but not affect the temporal causality.
>
> $\underline{\text{Equation (6)}}$ conforms to the above property: First, we use RoPE (indexed by $i, j$) on the temporal dimension to avoid permutation-invariance (we empirically highlight the advantage of RoPE over other embeddings in Section 4.5). Second, $u$ and $v$ (judged by $m, n$) are used to distinguish diagonal elements (endogenous variables) from non-diagonal elements (exogenous variables). The permutation-equivariance of variables can be satisfied (this work [1] provides a detailed proof).
>
> Therefore, $\underline{\text{Equation (6)}}$ is unaffected if the variable order of the flattened sequence is changed by $\underline{\text{Equation (4) and (5)}}$. **The variant of variable order actually affects the formulation of the mask** in $\underline{\text{Equation (8)}}$. In our paper, we use the **temporal-first** approach (i.e., two series are divided into six patches indexed by A: 1 2 3, B: 4 5 6, as shown in $\underline{\text{Figure 2}}$), the mask is formulated as $\mathcal{C}\otimes\mathcal{T}$. Invertedly, if we use the **variable-first approach** (A: 1 3 5, B: 2 4 6), the mask can be proved to be $\mathcal{T}\otimes\mathcal{C}$.
>
> **Q2**: Comparison with Moirai in the zero-shot scenario.
>
> Thanks for your valuable suggestions. We provided **a comprehensive zero-shot evaluation with more large time series models including different sizes of Moirai**. Please refer to $\underline{\text{Table 6}}$ in the revision for the details.
>
> [1] Zaheer et al. Deep Sets.

---

> ### Author Response · Authors · 2024-11-21
> **Summary of Response to Reviewer V4mu**
>
> Thank you again for your comments, which are very helpful for us to improve the quality of the paper. We **explored the properties of TimeAttention and enlarged zero-shot evaluations**. If you have any further questions, we are looking forward to discussing with you.

---

> ### Author Response · Authors · 2024-11-25
> **Eagerly Await Your Response**
>
> Dear Reviewer V4mu,
>
> Thanks again for your valuable and constructive review. After providing sufficient evidence and faithfully adding the requested analysis, two of the reviewers have raised the score. However, we find that the overall rating is still on the marginal threshold. We kindly hope you can consider our response and the conversation with other reviewers. If there are any further comments from your side, we will be happy to address them before the rebuttal period ends. If there are none, then we would appreciate it if you could reconsider your rating.
>
> Thank you once again for your support and constructive feedback.
>
> All the best,
>
> Authors

---

> > ### Comment · Reviewer_V4mu · 2024-11-27
> > **Official Comment by Reviewer V4mu**
> >
> > Thank you for your detailed and thorough response to the concerns raised. Your clarifications and additional experiments have addressed my concerns, and I keep my score.

---

> ### Author Response · Authors · 2024-11-27
> **Thanks for Your Positive Feedback**
>
> Dear Reviewer V4mu,
>
> We sincerely appreciate your valuable review and feedback. We are also glad to hear that our clarifications and additional experiments have addressed your concerns.
>
> Given the borderline decision, we would like to kindly inquire about what additional information or changes would be necessary to further enhance the overall quality of our paper.
>
> During the discussion phase, we followed the constructive comments of all reviewers, which is reflected in the revised version of **24 pages** and new **100+ experiments**, including:
>
> * **Empirical studies of the long-context forecasting**: Given that the long-context bottleneck received less attention in the community, we reveal the performance degradation of prevalent models on well-acknowledged datasets (Figure 3) and real-world applications (Table 13) with in-depth representation analysis (Figure 11). The explorations keep to the principles of forecasting and highlight the potential of Generative Transformers, which may provide a distinct architectural choice for future deep time series models.
> * **A better illustration of our technical novelty**: Based on the underexploited causal implementation of multivariate forecasting on Transformers, we provide an intuitive example (Figure 12) in the revision to highlight the challenges of capturing dependencies between grouped patch tokens. We revised the overall writing of the paper and better clarified the disentangling perspective underlying TimeAttention and its properties adept to multivariate time series.
> * **Extensive baseline models and comprehensive benchmarks**: In addition to the zero-shot evaluation based on your valuable suggestion, we provide comprehensive evaluations (please refer to Q1 of Reviewer MkP7) with state-of-the-art non-Transformer models, covariate-informed models, and LLM-based forecasters, further validating that Timer-XL can be a unified backbone for different forecasting tasks.
>
> **If there are any further comments from your side, we are open to extending our evaluation and paper even further**. Thank you once again for your support and the productive discussion.
>
> All the best,
>
> Authors

---

### Official Review · Reviewer_MkP7 · 2024-10-31

**Soundness:** 2
**Presentation:** 2
**Contribution:** 2
**Rating:** 5
**Confidence:** 4

**Summary:**

This paper proposes Timer-XL, a generative Transformer model for uni & mul-variate time series forecasting, which addresses the limitation of short context lengths in existing time series Transformers and allows the model to capture both intra- and inter-series dependencies. Specifically, the authors design TimeAttention, which incorporates relative position embeddings and causal masking to effectively learn temporal dependencies. Evaluations across various benchmarks demonstrate that Timer-XL achieves state-of-the-art performance in univariate, multivariate, and covariate-informed forecasting tasks.

**Strengths:**

1. The paper proposes multivariate next token prediction for time series forecasting. This paradigm unifies univariate, multivariate, and covariate-informed forecasting, by treating them as a long-context generation problem.
2. This paper introduces TimeAttention, a novel self-attention mechanism for time series data. TimeAttention captures fine-grained intra and inter-series dependencies, preserves causality in forecasting, and incorporates position embeddings.
3. Experiments show that extending the context length generally improves accuracy, highlighting the context bottleneck issue.

**Weaknesses:**

1. This paper focuses heavily on comparing Timer-XL with other Transformer models, particularly PatchTST, Timer, and lacks a broader comparison with other non-Transformer time series forecasting models. Also, how is Timer-XL compared with some recent LLM-based models? e.g., [1], [2].
2. This paper doesn't extensively discuss the computational cost of Timer-XL. Though it provides a theoretical derivation, a more detailed analysis of the computational resources required, especially when handling high-dimensional time series with long contexts. E.g., how about the number of parameters or training/inference time of Timer-XL as compared with existing solutions.
3. The extension from univriate to multivariate seems to be an over-simplied way, in that using RoPE to as a positional embedding. Take traffic data as an example, how can RoPE reflect the spatial correlations among traffic network?
4. The authors emphasize long context for time series forecasting, however, for some domains, it may not be necessary for such a long context, e.g., traffic data with periodicities. This is also shown in Figure 3.  Also, one can define a large patch token (more time points). In this way, can the context length also be shortened?

[1] Time-LLM: Time Series Forecasting by Reprogramming Large Language Models, ICLR 2024

[2] Large Language Models Are Zero-Shot Time Series Forecasters, NeurIPS 2023

**Questions:**

Please see the weaknesses.
Also, it will be good if the authors can use a figure to illustrate patch token for univariate and multivariate data.

---

> ### Author Response · Authors · 2024-11-21
> **Response to Reviewer MkP7 (Part 1)**
>
> Many thanks to Reviewer MkP7 for providing a valuable review.
>
> **Q1**: Comparison with LLM-based and non-Transformers forecasters.
>
> Thanks for your valuable suggestions, we **extensively included state-of-the-art methods** to enlarge our forecasting baselines in $\underline{\text{Table 3}}$: Time-LLM[1], LLMTime[2], TimeMixer[3], and FiTS[4]. Here are the results:
>
> | ETTh1 (MSE\|MAE) | TimeMixer          | FiTS               | Time-LLM       | LLMTime        | Timer-XL           |
> | ---------------- | ------------------ | ------------------ | -------------- | -------------- | ------------------ |
> | Pred-96          | 0.381 \| 0.408     | 0.378 \| 0.401     | 0.380 \| 0.412 | 0.451 \| 0.519 | **0.364 \| 0.397** |
> | Pred-192         | 0.416 \| 0.430     | 0.410 \| **0.420** | 0.408 \| 0.431 | 0.503 \| 0.528 | **0.405** \| 0.424 |
> | Pred-336         | **0.427** \| 0.434 | 0.431 \| **0.433** | 0.435 \| 0.443 | 0.537 \| 0.553 | **0.427** \| 0.439 |
> | Pred-720         | 0.453 \| 0.457     | 0.441 \|**0.451**  | 0.444 \| 0.463 | 0.573 \| 0.588 | **0.439** \| 0.459 |
>
> | ECL (MSE\|MAE) | TimeMixer      | FiTS           | Time-LLM       | LLMTime        | Timer-XL           |
> | -------------- | -------------- | -------------- | -------------- | -------------- | ------------------ |
> | Pred-96        | 0.132 \| 0.228 | 0.142 \| 0.244 | 0.137 \| 0.244 | 0.187 \| 0.284 | **0.127 \| 0.219** |
> | Pred-192       | 0.149 \| 0.244 | 0.156 \| 0.256 | 0.158 \| 0.266 | 0.204 \| 0.301 | **0.145 \| 0.236** |
> | Pred-336       | 0.166 \| 0.262 | 0.173 \| 0.272 | 0.183 \| 0.292 | 0.228 \| 0.328 | **0.159 \| 0.252** |
> | Pred-720       | 0.206 \| 0.298 | 0.213 \| 0.304 | 0.247 \| 0.348 | 0.279 \| 0.385 | **0.187 \| 0.277** |
>
> | Traffic (MSE\|MAE) | TimeMixer      | FiTS           | Time-LLM       | LLMTime        | Timer-XL           |
> | ------------------ | -------------- | -------------- | -------------- | -------------- | ------------------ |
> | Pred-96            | 0.362 \| 0.259 | 0.390 \| 0.275 | 0.376 \| 0.280 | 0.518 \| 0.406 | **0.340 \| 0.238** |
> | Pred-192           | 0.379 \| 0.267 | 0.402 \| 0.279 | 0.397 \| 0.294 | 0.546 \| 0.422 | **0.360 \| 0.247** |
> | Pred-336           | 0.394 \| 0.276 | 0.415 \| 0.285 | 0.420 \| 0.311 | 0.581 \| 0.462 | **0.377 \| 0.256** |
> | Pred-720           | 0.432 \| 0.299 | 0.453 \| 0.306 | 0.448 \| 0.326 | 0.656 \| 0.524 | **0.418 \| 0.279** |
>
> | Weather (MAE\|MAE) | TimeMixer      | FiTS               | Time-LLM    | LLMTime        | Timer-XL           |
> | ------------------ | -------------- | ------------------ | ----------- | -------------- | ------------------ |
> | Pred-96            | 0.151 \| 0.207 | **0.145 \| 0.197** | 0.149 0.200 | 0.222 \| 0.309 | 0.157 \| 0.205     |
> | Pred-192           | 0.198 \| 0.249 | **0.186 \| 0.237** | 0.193 0.243 | 0.310 \| 0.377 | 0.206\|  0.250     |
> | Pred-336           | 0.247 \| 0.287 | **0.241 \| 0.277** | 0.243 0.284 | 0.366 \| 0.436 | 0.259 \| 0.291     |
> | Pred-720           | 0.344 \| 0.346 | 0.342 \| 0.347     | 0.315 0.336 | 0.459 \| 0.498 | **0.337 \| 0.344** |
>
> | Solar. (MSE\|MAE) | TimeMixer      | FiTS           | Time-LLM       | LLMTime        | Timer-XL            |
> | ----------------- | -------------- | -------------- | -------------- | -------------- | ------------------- |
> | Pred-96           | 0.187 \| 0.254 | 0.197 \| 0.248 | 0.224 \| 0.289 | 0.271 \| 0.355 | **0.162 \|  0.221** |
> | Pred-192          | 0.205 \| 0.263 | 0.218 \| 0.258 | 0.248 \| 0.315 | 0.290 \| 0.370 | **0.187 \| 0.239**  |
> | Pred-336          | 0.220 \| 0.270 | 0.239 \| 0.270 | 0.269 \| 0.338 | 0.318 \| 0.399 | **0.205 \| 0.255**  |
> | Pred-720          | 0.240 \| 0.285 | 0.249 \| 0.278 | 0.310 \| 0.396 | 0.341 \| 0.434 | **0.238 \|0.279**   |
>
> Overall, our model achieves the best performance under **78%** settings.
>
> **Q2**: More discussion of the computational cost of Timer-XL.
>
> Thanks for your scientific rigor. We **included quantitive results of the computational cost** on high-dimensional time series (ECL: 321 variables) with a long lookback length (672), leading to the context of 200k+ time points. We use the same model hyperparameters according to the configurations of $\underline{\text{Table 10}}$:
>
> | Model    | Training Speed (s/iter) | Inference Speed (s/iter) | Params (M) | Performance (MSE) |
> | -------- | ----------------------- | ------------------------ | ---------- | ----------------- |
> | Moirai   | 0.2790  | 0.1128   | 15.86      | 0.163             |
> | PatchTST | 0.0719  | 0.0175  | 16.21      | 0.169             |
> | Timer-XL | 0.2480  | 0.0993      | 15.86      | 0.155             |
> | Timer    | 0.0527  | 0.0175      | 15.86      | 0.161             |
>
>
> Results are consistent with our theoretical analysis: Under the same hyperparameters, parameter counts are almost the same, while the multiplier of flops is less than the number of variables ($N$). We will also take it as essential work to improve the model efficiency of channel-dependent models.

---

> ### Author Response · Authors · 2024-11-21
> **Response to Reviewer MkP7 (Part 2)**
>
> **Q3**: Reclarification of the contribution of extending univariate to multivariate forecasting.
>
> Although the **formulation** to generalize from 1D sequences to 2D time series is straightforward, there are several unsolved challenges before our work:
>
> * **Capture Cross-Variable Dependencies**: Different from previous works that regard the whole variable as a token (iTransformer and TimeXer), Timer-XL captures **fine-grained dependencies between all variables in the patch level**. Thus, variable correlations can be **reflected by the fine-grained attention map**. For example, we show the correlations of the Traffic variables that correspond to the **aggregated attention map** in $\underline{\text{Figure 7}}$. The fine-grained attention map can also reflect the ACF and time-unaligned dependencies.
> * **Maintain Temporal Causality**: Timer-XL is built on decoder-only architecture, an underexploited backbone among current time series models. While encoder-only models are prevalently adopted, our empirical results reveal that keeping the causality alleviates long-context degradation better ($\underline{\text{Figure 3}}$) and boosts the performance ($\underline{\text{Table 2-5}}$). Ablation study ($\underline{\text{Table 12}}$) that analyzes different positional embeddings is also a pioneer work.
> * **Permutation-Equivariance of Variables**: To extend univariate to multivariate data, it is crucial to ensure the positional equivalence of variables. Formally, $\operatorname{Attn}(\mathcal{P}(V1,...,VN)) = \mathcal{P}(\operatorname{Attn}(V1, ..., VN))$, where $\mathcal{P}$ is any permutation of $N$ variables. Therefore, TimeAttention is **not a trivial utilization of RoPE** but also satisfies the desired property: $u$ and $v$ are adopted to distinguish diagonal elements (endogenous variables) from non-diagonal elements (exogenous variables). In this way, the permutation-equivariance of variables can be satisfied (this work [5] provides detailed proof).
>
> To our best knowledge, the paradigm of next token prediction is still unexplored in task-specific forecasters, let alone extend to multiple forecasting tasks. Without changing the self-attention, we design a masking mechanism to **disentangle** the large attention map into formalizable and interpretable temporal and variable dependencies. In this perspective, our implementation is **simple but effective**. In summary, here is a comparison with previous time-series Transformers:
>
> | Time-Series Transformers                   | iTransformer | PatchTST | Moirai/Crossformer | Timer | Timer-XL |
> | ------------------------------------------ | ------------ | -------- | ------------------ | ----- | -------- |
> | Generative (Causality, Context-Flexibility) | No           | No       | No                 | Yes   | **Yes**  |
> | Patch-Level Dependencies                   | No           | Yes      | Yes                | Yes   | **Yes**  |
> | Cross-Variable Modeling                    | Yes          | No       | Yes                | No    | **Yes**  |
>
> **Q4**: Motivations of enlarging the context length.
>
> We agree with your opinion that long contexts would not always be beneficial. In fact, we want to draw the attention of the community to the **long-context bottleneck** instead of advocating for everyone to adopt long contexts. In addition to $\underline{\text{Figure 3}}$, we provided a new case study here :
>
> | **MSE of ERA5 Dataset (Context-Length)** | **Timer-XL** | **PatchTST** | **DLinear** |
> | ---------------------------------------- | ------------ | ------------ | ----------- |
> | Lookback-8 (1 Day)    | 0.0847       | 0.0897       | 0.0970      |
> | Lookback-32 (4 Day)     | 0.0713       | 0.0778       | 0.0841      |
> | Lookback-56 (1 Week)     | 0.0688       | 0.0785       | 0.0814      |
> | Lookback-224 (1 Month)     | 0.0675       | 0.0745       | 0.0788      |
> | Lookback-960 (4 Month)   | 0.0667       | 0.1194       | 0.0773      |
> | Lookback-2944 (1 Year)    | 0.0663       | 0.1109       | 0.0763      |
>
> In the meteorological domain, it is of great demand to support the context of more than years to contain a specific cycle (such as El Nino). More importantly, it draws similar conclusions that **encoder-only Transformers encounter performance degradation in long context series** (even inferior to the linear model at last), which can be concealed due to the short context adopted in previous benchmarks. Although Timer-XL with the longest context is still worse than using a searched one in $\underline{\text{Figure 3}}$, it is more a matter of feature engineering. We focus more on the architecture, with the results revealing **Generative Transformers can better cope with long contexts**. By our visualization of attention in encoder-/decoder-only Transformers ($\underline{\text{Figure 11}}$ in the revised version), **the decoder-only model can selectively focus on long contexts**, while attention maps of PatchTST are oversmooth and make the model wrongly focus on the noisy part.

---

> ### Author Response · Authors · 2024-11-21
> **Response to Reviewer MkP7 (Part 3)**
>
> Another inspiring fact is that Generative Transformers have flexibility on the input length (as long as the input length during inference is shorter than the context length during training). It leaves a promising application: Training a model with a long context and applying it with only partial length or being adapted to a specific length.
>
> Therefore, long context is a **basic model capability towards foundation models**, which can support emergence capabilities such as prompting, ICL, RAG, etc. However, the **long-context forecasting paradigm received less attention** in the current community, which can be due to the lack of benchmarks. Therefore, we spent a lot of time curating the ERA5 dataset (500 thousand time points with yearly context and thousands of variables), which serves as a contribution that cannot be ignored.
>
> We appreciate your suggestion about using a large patch size, which can shorten the context length and reduce steps of autoregression. We **provided the results using a large patch size** in the middle of $\underline{\text{Figure 8}}$. The finding is noteworthy and indicates potential improvement in future works.
>
> **Q5**: Illustration of the patch token for univariate and multivariate data.
>
> Thanks for your valuable suggestions. For a better illustration, we provided $\underline{\text{Figure 12}}$ in the revision, which highlighted the challenges and our pipeline of capturing fine-grained dependencies between patch tokens.
>
> [1] Time-LLM: Time Series Forecasting by Reprogramming Large Language Models. ICLR 2024.
>
> [2] Large Language Models Are Zero-Shot Time Series Forecasters. NeurIPS 2023.
>
> [3] FITS: Modeling Time Series with 10k Parameters. ICLR 2024.
>
> [4] TimeMixer: Decomposable Multiscale Mixing for Time Series Forecasting. ICLR 2024.
>
> [5] Zaheer et al. Deep Sets.

---

> ### Author Response · Authors · 2024-11-21
> **Summary of Response to Reviewer MkP7**
>
> Thank you again for your comments, which are very helpful for us to improve the quality of the paper. We **provided comprehensive benchmarks and clarified more on the technical design and long-context bottlenecks**. Please reconsider the efforts and contributions of our work. If you have any further questions, we are looking forward to discussing with you.

---

### Official Review · Reviewer_r6jB · 2024-11-02

**Soundness:** 2
**Presentation:** 2
**Contribution:** 2
**Rating:** 6
**Confidence:** 5

**Summary:**

The paper proposes Timer-XL, a transformer decoder model for time series forecasting. Building upon the existing Timer model, Timer-XL extends the model with a longer context and a masking-based approach called TimeAttention to handle multivariate/covariate scenarios. Empirical results have been reported on univariate, multivariate and covariate experiments on some benchmark datasets.

**Strengths:**

a) The paper studies an interesting problem of long context modeling in the context of time series forecasting. Authors attempt to connect long context to scenarios beyond univariate modeling through next-token style modeling of multivariate and covariate-informed time series.

b) Experiments have been conducted on many diverse settings although the experiments themselves have some limitations.

**Weaknesses:**

a) While the problem of long context modeling is interesting, the primary weakness of this work is the lack of clarity about the goal and a proper scope. The discussion is confusing and often only loosely relates to the long context setting which appears to be the primary goal. Authors claim that "existing transformers in the time series field crucially encounter the context bottleneck" which is not as critical of a problem as being portrayed here. Such claims require serious empirical justification which is missing from the paper (experiment 1 does not go too far, see below). In reality, long context scenarios may be helpful but _mostly_ in specific high-frequency scenarios. Consider a 5min granularity time series with weekly seasonal behavior. One would need a context larger than 2K to understand the seasonal behavior from the time series history. Such cases should have been better highlighted to justify the central claim of the paper. That said, long context univariate modeling may not always yield improvements. It may also worsen the accuracy, e.g., in the case of distribution shifts.

Coming to the utility of long context for multivariate/covariate-informed forecasting, this has been studied (although not explicitly) in Moirai. While the explicit perspective in this paper is interesting, the primary problem is not that one can do multivariate/covariate modeling through long context but that one needs to be able to do it in a zero-shot sense, as attempted in Moirai. As per my understanding, the settings being studied here are task-specific and not a single pretrained Timer-XL model being used for all experiments. Please correct me if I am wrong.

b) The technical novelty of this work is limited in light of works such as Moirai. "TimeAttention" is a masking scheme that extends causal univariate patch-based modeling to the multivariate setting.

c) The empirical results lack comprehensiveness and are not particularly strong.

**Experiment 4.1**: The benchmark is fairly small to draw conclusions. Even in these scenarios, the benchmark shows that long context yields to diminishing returns (or worse performance) beyond a point, 1 month in most cases. 1 month @ 1h granularity amounts to a context length of 720 which is not far off from the context length of many time series models (Chronos, Moirai, TimesFM, etc.). Furthermore, I don't understand why PatchTST is the only baseline being studied here. I also don't quite understand what's unique about Timer-XL here that yields better performance over PatchTST. Could it be the larger number of parameters?

Minor: The analysis on normalization does not belong to this section and brings limited value to the discussion. Normalization mostly helps when time series in a dataset have drastically different scales. Models working fine without normalization for a single task of land-surface temperature forecasting is not a surprising result.

**Experiment 4.2**: The results in Table 2 and 4 are not strong when compared to pretrained baselines such as Moirai. Why is Moirai missing from Fig 4?

Minor: DeepAR and N-BEATS are not numerical simulation based methods.

**Experiment 4.3**: More baselines that can incorporate covariates are needed before conclusions can be drawn. For example, you can consider adding N-BEATSx, NHITS, DeepAR, etc. GluonTS also provides an implementation of PatchTST which can incorporate covariates.

**Experiment 4.4**: The benchmark selected in this experiment for out of domain generalization is severely limited. 5/7 datasets belong to the same domain and with 4 being essentially the same dataset (ETTh1, ETTh2, ETTm1, ETTm2). This is not enough to draw conclusions about a "pretrained model". Please check the benchmarks in the Chronos or TimesFM papers.

d) The model only enables point predictions. In forecasting, one is often interested in the entire distribution of future possibilities.

**Questions:**

See above.

- Which version of Moirai was used for the experiments? 1.0 or 1.1?

---

> ### Author Response · Authors · 2024-11-21
> **Response to Reviewer r6jB (Part 1)**
>
> Many thanks to Reviewer r6jB for providing a detailed and in-depth review, which helped us significantly improve the quality of our submission.
>
> **Q1**: Clarification of the position of our work.
>
> Thank you very much for providing such an essential suggestion. Our work aims for a **unified backbone for one-for-all forecasting**. Our evaluations are divided into two parts: **supervised training** on Timer-XL for different tasks  (Sections 4.1, 4.2, and 4.3) and **large-scale pre-training on long-context time series** (Section 4.4).
>
> We apologize for the potential misleading caused by several large models in the supervised training baseline, such as Moirai. Except for Section 4.4, Timer-XL, Moirai, and other baseline models are **trained from scratch**. Therefore, we **revised the name of Moirai to the more appropriate UniTST[1]** (w/o dispatchers) in these sections, a pioneer encoder-only backbone supporting unified forecasting. In Section 4.4, we intend to validate that long-context pre-training empowers large time series models (we appreciate your suggestion to use more comprehensive benchmarks, which we resolved in $\underline{\text{the response to Q4}}$).
>
> Your suggestion helped us better clarify the scope of our work, and **we revised the overall writing of the paper**.
>
> **Q2**: Empirical justification to reveal the context bottleneck.
>
> We agree with your opinion that enlarging contexts would not always be beneficial, which needs a better empirical justification. As per your request, we **provided more practical cases of long-context forecasting** where using long contexts is an essential requirement: In the meteorological ERA5, despite its low sampling frequency (3 hours), it is necessary to support the context of more than years to contain a specific cycle (such as El Nino):
>
> | **MSE of ERA5 Dataset (Context-Length)** | **Timer-XL** | **PatchTST** | **DLinear** |
> | ----------------------------------------- | ------------ | ------------ | ----------- |
> | Lookback-8 (1 Day)                        | 0.0847       | 0.0897       | 0.0970      |
> | Lookback-32 (4 Day)                       | 0.0713       | 0.0778       | 0.0841      |
> | Lookback-56 (1 Week)                      | 0.0688       | 0.0785       | 0.0814      |
> | Lookback-224 (1 Month)                    | 0.0675       | 0.0745       | 0.0788      |
> | Lookback-960 (4 Month)                    | 0.0667       | 0.1194       | 0.0773      |
> | Lookback-2944 (1 Year)                    | 0.0663       | 0.1109       | 0.0763      |
>
> The performance of Timer-XL and DLinear generally improves with the increased context length. It also reveals the performance degradation of PatchTST. Similar to the observations in $\underline{\text{Figure 3}}$, **encoder-only Transformers encounter the degradation earlier, which can be concealed due to the short context adopted on previous benchmarks**.
>
> Although PatchTST has conducted an initial exploration in the context of hundreds of time points, it is inferior in ever-long contexts, and we find that **Generative Transformers mitigate performance degradation well**. We further analyze this problem:
>
> * **Increased Supervision**: Generative Transformers predict all tokens, including tokens in the lookback window.
> * **Causality Maintainance**: Using causal masks leads to **coherent token embeddings**, while the unmasked attention in PatchTST breaks the causality and cannot tell these tokens.
>
> We **provided representation analysis** of encoder-/decoder-only Transformers in $\underline{\text{Figure 11}}$ of the revised version, revealing that **the decoder-only model can selectively focus on the long context while PatchTST may wrongly focus on noisy parts**.
>
> In addition to the empirical justification, we notice that recent large time series models also paid special attention to longer context length[2]\[3]. This capability serves as the basis for supporting emergent capabilities such as prompting, ICL, RAG, etc. Therefore, we believe that context bottlenecks deserve further exploration in this community.

---

> ### Author Response · Authors · 2024-11-21
> **Response to Reviewer r6jB (Part 2)**
>
> **Q3**: Reclarification of the technical novelty of our work.
>
> Thanks for your valuable feedback regarding the novelty. Our contribution lies in two aspects:
>
> **1. Generative Transformers for Long-Context Series**: As you mentioned, long-context Transformers for multivariate modeling have been studied. Moirai and UniTST are all encoder-only models. Our difference lies in **the implementation of Generative Transformers**. Motivated by the success of GPTs in NLP, which successfully accommodates **millions of tokens**, our work validated that they can better solve performance degradation in long-context time series.
>
> To our best knowledge, Generative Transformers are still underexploited among task-specific models, let alone the implementation for multivariate series. In this perspective, our technical novelty is not limited and does fill in the blanks.
>
> Although the longest context may be inferior to a searched one ($\underline{\text{Figure 3}}$), it is more a matter of feature engineering, and searching also requires heavy time and resources. A longer lookback does introduce irrelevant noise and distribution shift. Therefore, finding an architecture that can **adaptively select information in vast observations** is significant, which highlights our contribution of adopting Generative Transformers on long-context univariate forecasting.
>
> **2. TimeAttention for Unified Forecasting**: The equation of calculating attention is inspired by Moirai. We greatly appreciate the pivot work with the explicit citation. However, we introduce the masking formulation, whose key lies in the **grouped causality of flattened 2D sequences**. We derive it based on the Kronecker Product, which can **disentangle the large attention map** (as shown in $\underline{\text{Figure 7}}$) into explicit temporal and variable dependencies. It can generalize to pre-defined correlations (such as covariates), which is unexplored in previous works.
>
> To sum up, we explore the potential of Generative Transformers for (1) long-context univariate series and (2) multiple time series. Technically, we provide a decoupling perspective of multivariate next token prediction and implement it on the decoder-only architecture, which may inspire future explorations.
>
> **Q4**: Further explanations of experiments in our work.
>
> We appreciate your questions regarding the experiments. We actually spend a lot of time on these experiments. Please allow us to clarify the experiments you mentioned in detail:
>
> * **Experiment 4.1 - Long-Context Univairtae Forecasting**: Even though we only compared two models here, **they are representative and work exactly for our studied problem**: Which architecture of Transformer can better capture temporally long dependencies? In this univariate setting, Timer-XL is degraded into the decoder-only Timer, and the corresponding counterpart is encoder-only PatchTST (UniTST/iTransformer will be regarded as PatchTST/FFN layers). Further, we **added DLinear** based on your inspiration in the **revised $\underline{\text{Figure 3}}$**. For the fairness of comparison, please refer to $\underline{\text{Table 10}}$, where we keep the hyperparameters of Transformers all the same.
>
> * (Minor): Section 4.1 discusses instance normalization (ReVIN). Since ReVIN can generally improve the performance of the previous encoder-only Transformers but leads to special problems in Generative Transformers (e.g., unmatched statistics in multi-step autoregression), we provide more results indicating that using Timer-XL without ReVIN can still achieve competitive performance in these well-acknowledged benchmarks, while the performance of PatchTST may heavily rely on this normalization:
>
>   | Dataset-672-Pred-96 (MSE\|MAE) | Timer-XL with ReVIN | Timer-XL w/o ReVIN | PatchTST with ReVIN | PatchTST w/o ReVIN |
>   | ------------------------------ | ------------------- | ------------------ | ------------------- | ------------------ |
>   | ETTh1                          | 0.364 \| 0.397      | 0.370 \| 0.401     | 0.370 \| 0.399      | 0.421 \| 0.448     |
>   | Weather                        | 0.157 \| 0.205      | 0.151 \| 0.205     | 0.149 \| 0.198      | 0.173 \| 0.242     |
>   | ECL                            | 0.127 \| 0.219      | 0.130 \| 0.225     | 0.129 \| 0.222      | 0.138 \| 0.244     |
>
> * **Experiment 4.2 - Multi-Station Forecasting**: Since it takes a long time to train on GTWSF, we report the original results of baseline models from Corrformer[4]. According to your suggestion, we **added UniTST** to this benchmark and revised. $\underline{\text{Figure 4}}$.
>
>   | MSE         | UniTST | Timer-XL |
>   | ----------- | ------ | -------- |
>   | Global Temp | 7.356  | 7.172    |
>   | Global Wind | 3.807  | 3.786    |
>
> * (Minor): Thanks very much for pointing out the wrong labels! We fixed it in $\underline{\text{Figure 4}}$.

---

> ### Author Response · Authors · 2024-11-21
> **Response to Reviewer r6jB (Part 3)**
>
> * **Experiment 4.3 - Covariate-Informed Forecasting**: We reported the baseline results from TimeXer, where PatchTST and other models are adapted to incorporate covariates. Also, we **added new task-specific forecasters**:
>
>   | MSE\|MAE | **DeepAR**     | **N-BEATSX**   | **Timer-XL**   |
>   | -------- | -------------- | -------------- | -------------- |
>   | **NP**   | 0.277 \| 0.308 | 0.272 \| 0.301 | 0.234 \| 0.262 |
>   | **PJM**  | 0.121 \| 0.213 | 0.097 \| 0.189 | 0.089 \| 0.187 |
>   | **BE**   | 0.416 \| 0.279 | 0.389 \| 0.265 | 0.371 \| 0.243 |
>   | **FR**   | 0.432 \| 0.237 | 0.393 \| 0.211 | 0.381 \| 0.204 |
>   | **DE**   | 0.512 \| 0.458 | 0.499 \| 0.447 | 0.434 \| 0.415 |
>
> * **Experiment 4.4 - Zero-shot forecasting**: Since the benchmark of zero-shot forecasting is under rapid development. We **provided zero-shot results** on these datasets that are not included in UTSD during pre-training.
>
>   | Zero-shot (MSE\|MAE) | Timer-XL | Timer       |
>   | ----------------------------- | ----------- | ----------- |
>   | BDG-2 Panther [5]       | 0.331 \| 0.370 | 0.354 \| 0.395 |
>   | Cockatoo [5]               | 0.278 \| 0.377 | 0.324 \| 0.396 |
>   | GFC17 Load [5]              | 0.289 \| 0.377 | 0.340 \| 0.425 |
>   | Taxi New York Passengers [6] | 0.179 \| 0.302 | 0.524 \| 0.580 |
>   | Temps [6]                  | 0.790 \| 0.702 | 0.981 \| 0.782 |
>
>   Please feel free to let us know if you want us to evaluate the performance on more datasets. We also provided **zero-shot evaluation with more large models**. Please also refer to $\underline{\text{Table 6}}$  in our revision for the details.
>
> **Q5**: Support for probabilistic forecasting.
>
> It is necessary to support probabilistic forecasting because many task-specific models lack this capability. It is a direction worthy of specific study. We will explore the following direction: integrating proper distribution of time series and employing probabilistic prediction heads for heterogeneous variables. Your suggestion will guide us in refining our methodology.
>
> [1] UniTST: Effectively Modeling Inter-Series and Intra-Series Dependencies for Multivariate Time Series Forecasting.
>
> [2] In-Context Fine-Tuning for Time Series Foundation Models.
>
> [3] Time-MoE: Billion-Scale Time Series Foundation Models with Mixture of Experts.
>
> [4] Interpretable Weather Forecasting for Worldwide Stations with A Unified Deep Model.
>
> [5] Unified Training of Universal Time Series Forecasting Transformers.
>
> [6] A Decoder-Only Foundation Model for Time Series Forecasting.

---

> ### Author Response · Authors · 2024-11-21
> **Summary of Response to Reviewer r6jB**
>
> Thank you again for your detailed comments, which are very helpful for us to improve the quality of the paper. We **clarified the positioning and contribution more rigorously, addressed the concerns, and enhanced the comprehensiveness of experiments**. Please reconsider our contributions and insights from our work. If you have any further questions, we are looking forward to discussing with you.

---

### Author Response · Authors · 2024-11-21
**Summary of Revisions**

We sincerely thank all the reviewers for their insightful reviews and valuable comments, which are instructive for us to improve our paper.

In this work, we propose **a unified backbone for one-for-all forecasting**, including one model for varying forecasting horizons (based on Generative Transformers) and one architecture for different forecasting tasks (based on causal self-attention for multivariate time series). Insightfully, we find that **Generative Transformers can better tackle long-context time series**, while previous deep models generally fail. Empirical results include **comparison with state-of-the-art task-specific models** and **zero-shot evaluation by pre-training on billion-level time series points**.

We're pleased that the reviewers agree our paper "**highlights the context bottleneck issue**" (Reviewer r6jB, MkP7), which we further delved into in this revision, "**demonstrates a high level of completeness with promising empirical evaluation**" (Reviewer V4mu), and "**the proposed self-attention is novel**" (Reviewer MkP7, V4mu).

The reviewers raised insightful and constructive concerns. We **spent nine days** addressing all the issues by providing sufficient evidence and requested results. Here is the summary of the major revisions:

* **Motivation and Scope  (Reviewer r6jB)**: We provided empirical evaluations to emphasize the motivation of long-context forecasting, reclarified the scope of our work, and revised the overall writing of the paper (extra **3 pages** compared with the initial submission).
* **Technical Contributions (Reviewer r6jB, MkP7)**: We clarified the challenges of adopting Generative Transformers for unified time series forecasting. Our implementation facilitates channel-dependent modeling in the flexible decoder-only architecture, an unexploited backbone in this community.
* **Clarification of Experiment Details  (Reviewer r6jB)**: To improve the clarity, we provided the concerned experimental configurations and eliminated potential misleading in the establishment of baseline models.
* **Comprehensive Evaluations (Reviewer r6jB, MkP7, V4mu)**: We included all requested evaluations (100+ experiments in total), including new task-specific baselines, zero-shot benchmarks based on large-scale pre-training, computational cost statistics, and long-context representation analysis.

All updates are highlighted in blue $\underline{\text{in the revised paper}}$. The valuable suggestions from reviewers are very helpful for us to revise the paper in a better shape. We hope our response has fulfilled the reviewer's expectations and would be very happy to answer any further questions.

---

### Meta-Review · Area_Chair_CXsX · 2024-12-23

**Metareview:**

This paper introduces a generative transformer model for unified timeseries forecasting. They propose a new form of TimeAttention which allows for the application of generative transformers on multivariate data and demonstrate improvements in long-context regimes. They provide results with both supervised training as well as unified pretraining across many datasets.

The reviewers all agreed that the paper addresses an important and timely topic, and that the work has a number of noteworthy contributions. However, the reviewers also noted a few weaknesses in the initial submission, including limited novelty over approaches like Morai and that the move from univariate to multivariate data is handled in an oversimplistic manner. Reviewers also brought up some concerns around the motivation of long context and whether the experiments adequately demonstrated that this improvement is helpful in practical settings.

During the rebuttal phase, the authors did an excellent job of responding to these concerns, and further clarified the novelty of their work.

**Additional Comments On Reviewer Discussion:**

To address Reviewers r6jB and MkP7 concerns about needing more empirical results to clarify the motivation of long-context forecasting, the authors added additional evaluations on a meteorological dataset, the ERA5 Dataset, where they show that long context forecasting can be beneficial. In response to Reviewers r6jB and V4mu who requested a zero-shot benchmark, the authors provide new results which highlight that Timer-XL outperforms state-of-the-art models (such as Moirai) on zero-shot tasks. Finally, Reviewer MkP7 and r6jB requested additional baseline models and the authors conducted additional experiments to compare with a number of state-of-the-art non-Transformer models, covariate-informed models, and LLM-based forecasters (resulting in 100+ new experiments).

---

### Decision · Program_Chairs · 2025-01-22

Accept (Poster)